# Sample and Computationally Efficient Learning Algorithms under S-Concave Distributions

**Maria-Florina Balcan**
Machine Learning Department
Carnegie Mellon University, USA
ninamf@cs.cmu.edu

**Hongyang Zhang**[*]
Machine Learning Department
Carnegie Mellon University, USA
hongyanz@cs.cmu.edu

## Abstract

We provide new results for noise-tolerant and sample-efficient learning algorithms under $s$-concave distributions. The new class of $s$-concave distributions is a broad and natural generalization of log-concavity, and includes many important additional distributions, e.g., the Pareto distribution and $t$-distribution. This class has been studied in the context of efficient sampling, integration, and optimization, but much remains unknown about the geometry of this class of distributions and their applications in the context of learning. The challenge is that unlike the commonly used distributions in learning (uniform or more generally log-concave distributions), this broader class is not closed under the marginalization operator and many such distributions are fat-tailed. In this work, we introduce new convex geometry tools to study the properties of $s$-concave distributions and use these properties to provide bounds on quantities of interest to learning including the probability of disagreement between two halfspaces, disagreement outside a band, and the disagreement coefficient. We use these results to significantly generalize prior results for margin-based active learning, disagreement-based active learning, and passive learning of intersections of halfspaces. Our analysis of geometric properties of $s$-concave distributions might be of independent interest to optimization more broadly.

## 1 Introduction

Developing provable learning algorithms is one of the central challenges in learning theory. The study of such algorithms has led to significant advances in both the theory and practice of passive and active learning. In the passive learning model, the learning algorithm has access to a set of labeled examples sampled i.i.d. from some unknown distribution over the instance space and labeled according to some underlying target function. In the active learning model, however, the algorithm can access unlabeled examples and request labels of its own choice, and the goal is to learn the target function with significantly fewer labels. In this work, we study both learning models in the case where the underlying distribution belongs to the class of $s$-concave distributions.

Prior work on noise-tolerant and sample-efficient algorithms mostly relies on the assumption that the distribution over the instance space is log-concave [1, 12, 7, 30]. A distribution is *log-concave* if the logarithm of its density is a concave function. The assumption of log-concavity has been made for a few purposes: for computational efficiency reasons and for sample efficiency reasons. For computational efficiency reasons, it was made to obtain a noise-tolerant algorithm even for seemingly simple decision surfaces like linear separators. These simple algorithms exist for noiseless scenarios, e.g., via linear programming [28], but they are notoriously hard once we have noise [15, 25, 19]; This is why progress on noise-tolerant algorithms has focused on uniform [22, 26] and

---

[*]Corresponding author.

log-concave distributions [4]. Other concept spaces, like intersections of halfspaces, even have no computationally efficient algorithm in the noise-free settings that works under general distributions, but there has been nice progress under uniform and log-concave distributions [27]. For sample efficiency reasons, in the context of active learning, we need distributional assumptions in order to obtain label complexity improvements [16]. The most concrete and general class for which prior work obtains such improvements is when the marginal distribution over instance space satisfies log-concavity [32, 7]. In this work, we provide a broad generalization of all above results, showing how they extend to $s$-concave distributions ($s < 0$). A distribution with density $f(x)$ is *s-concave* if $f(x)^s$ is a concave function. We identify key properties of these distributions that allow us to simultaneously extend all above results.

**How general and important is the class of s-concave distributions?** The class of $s$-concave distributions is very broad and contains many well-known (classes of) distributions as special cases. For example, when $s \to 0$, $s$-concave distributions reduce to *log-concave* distributions. Furthermore, the $s$-concave class contains infinitely many fat-tailed distributions that do not belong to the class of log-concave distributions, e.g., Cauchy, Pareto, and $t$ distributions, which have been widely applied in the context of theoretical physics and economics, but much remains unknown about how the provable learning algorithms, such as active learning of halfspaces, perform under these realistic distributions. We also compare $s$-concave distributions with nearly-log-concave distributions, a slightly broader class of distributions than log-concavity. A distribution with density $f(x)$ is nearly-log-concave if for any $\lambda \in [0,1]$, $x_1, x_2 \in \mathbb{R}^n$, we have $f(\lambda x_1 + (1 - \lambda)x_2) \geq e^{-0.0154} f(x_1)^\lambda f(x_2)^{1-\lambda}$ [7]. The class of $s$-concave distributions includes many important extra distributions which do not belong to the nearly-log-concave distributions: a nearly-log-concave distribution must have sub-exponential tails (see Theorem 11, [7]), while the tail probability of an $s$-concave distribution might decay much slower (see Theorem 1 (6)). We also note that efficient sampling, integration and optimization algorithms for $s$-concave distributions have been well understood [13, 23]. Our analysis of $s$-concave distributions bridges these algorithms to the strong guarantees of noise-tolerant and sample-efficient learning algorithms.

## 1.1 Our Contributions

**Structural Results.** We study various geometric properties of $s$-concave distributions. These properties serve as the structural results for many provable learning algorithms, e.g., margin-based active learning [7], disagreement-based active learning [29, 21], learning intersections of halfspaces [27], etc. When $s \to 0$, our results exactly reduce to those for log-concave distributions [7, 2, 4]. Below, we state our structural results informally:

**Theorem 1** (Informal). *Let $\mathcal{D}$ be an isotropic $s$-concave distribution in $\mathbb{R}^n$. Then there exist **closed-form** functions $\gamma(s, m)$, $f_1(s, n)$, $f_2(s, n)$, $f_3(s, n)$, $f_4(s, n)$, and $f_5(s, n)$ such that*

1. *(Weakly Closed under Marginal) The marginal of $\mathcal{D}$ over $m$ arguments (or cumulative distribution function, CDF) is isotropic $\gamma(s, m)$-concave. (Theorems 3, 4)*
2. *(Lower Bound on Hyperplane Disagreement) For any two unit vectors $u$ and $v$ in $\mathbb{R}^n$, $f_1(s, n)\theta(u, v) \leq \Pr_{x \sim \mathcal{D}}[\text{sign}(u \cdot x) \neq \text{sign}(v \cdot x)]$, where $\theta(u, v)$ is the angle between $u$ and $v$. (Theorem 12)*
3. *(Probability of Band) There is a function $d(s, n)$ such that for any unit vector $w \in \mathbb{R}^n$ and any $0 < t \leq d(s, n)$, we have $f_2(s, n)t < \Pr_{x \sim \mathcal{D}}[|w \cdot x| \leq t] \leq f_3(s, n)t$. (Theorem 11)*
4. *(Disagreement outside Margin) For any absolute constant $c_1 > 0$ and any function $f(s, n)$, there exists a function $f_4(s, n) > 0$ such that $\Pr_{x \sim \mathcal{D}}[\text{sign}(u \cdot x) \neq \text{sign}(v \cdot x) \text{ and } |v \cdot x| \geq f_4(s, n)\theta(u, v)] \leq c_1 f(s, n)\theta(u, v)$. (Theorem 13)*
5. *(Variance in 1-D Direction) There is a function $d(s, n)$ such that for any unit vectors $u$ and $a$ in $\mathbb{R}^n$ such that $\|u - a\| \leq r$ and for any $0 < t \leq d(s, n)$, we have $\mathbb{E}_{x \sim \mathcal{D}_{u,t}}[(a \cdot x)^2] \leq f_5(s, n)(r^2 + t^2)$, where $\mathcal{D}_{u,t}$ is the conditional distribution of $\mathcal{D}$ over the set $\{x : |u \cdot x| \leq t\}$. (Theorem 14)*
6. *(Tail Probability) We have $\Pr[\|x\| > \sqrt{n}t] \leq \left[1 - \frac{cst}{1+ns}\right]^{(1+ns)/s}$. (Theorem 5)*

*If $s \to 0$ (i.e., the distribution is log-concave), then $\gamma(s, m) \to 0$ and the functions $f(s, n)$, $f_1(s, n)$, $f_2(s, n)$, $f_3(s, n)$, $f_4(s, n)$, $f_5(s, n)$, and $d(s, n)$ are all absolute constants.*

Table 1: Comparisons with prior distributions for margin-based active learning, disagreement-based active learning, and Baum's algorithm.

| | Prior Work | | Ours |
|---|---|---|---|
| Margin (Efficient, Noise) | uniform [3] | log-concave [4] | $s$-concave |
| Disagreement | uniform [20] | nearly-log-concave [7] | $s$-concave |
| Baum's | symmetric [9] | log-concave [27] | $s$-concave |

To prove Theorem 1, we introduce multiple new techniques, e.g., *extension of Prekopa-Leindler theorem* and *reduction to baseline function* (see the supplementary material for our techniques), which might be of independent interest to optimization more broadly.

**Margin Based Active Learning:** We apply our structural results to margin-based active learning of a halfspace $w^*$ under any isotropic $s$-concave distribution for both *realizable* and *adversarial* noise models. In the realizable case, the instance $X$ is drawn from an isotropic $s$-concave distribution and the label $Y = \text{sign}(w^* \cdot X)$. In the adversarial noise model, an adversary can corrupt any $\eta \ (\leq O(\epsilon))$ fraction of labels. For both cases, we show that there exists a *computationally efficient* algorithm that outputs a linear separator $w_T$ such that $\text{Pr}_{x \sim \mathcal{D}}[\text{sign}(w_T \cdot x) \neq \text{sign}(w^* \cdot x)] \leq \epsilon$ (see Theorems 15 and 16). The label complexity w.r.t. $1/\epsilon$ improves exponentially over the passive learning scenario under $s$-concave distributions, though the underlying distribution might be fat-tailed. To the best of our knowledge, this is the first result concerning the *computationally-efficient, noise-tolerant* margin-based active learning *under the broader class of $s$-concave distributions*. Our work solves an open problem proposed by Awasthi et al. [4] about exploring wider classes of distributions for provable active learning algorithms.

**Disagreement Based Active Learning:** We apply our results to agnostic disagreement-based active learning under $s$-concave distributions. The key to the analysis is estimating the disagreement coefficient, a distribution-dependent measure of complexity that is used to analyze certain types of active learning algorithms, e.g., the CAL [14] and $A^2$ algorithm [5]. We work out the disagreement coefficient under isotropic $s$-concave distribution (see Theorem 17). By composing it with the existing work on active learning [17], we obtain a bound on label complexity under the class of $s$-concave distributions. As far as we are aware, this is the first result concerning disagreement-based active learning that goes beyond log-concave distributions. Our bounds on the disagreement coefficient match the best known results for the much less general case of log-concave distributions [7]; Furthermore, they apply to the $s$-concave case where we allow an arbitrary number of discontinuities, a case not captured by [18].

**Learning Intersections of Halfspaces:** Baum's algorithm is one of the most famous algorithms for learning the intersections of halfspaces. The algorithm was first proposed by Baum [9] under symmetric distribution, and later extended to log-concave distribution by Klivans et al. [27] as these distributions are almost symmetric. In this paper, we show that approximate symmetry also holds for the case of $s$-concave distributions. With this, we work out the label complexity of Baum's algorithm under the broader class of $s$-concave distributions (see Theorem 18), and advance the state-of-the-art results (see Table 1).

We provide lower bounds to partially show the tightness of our analysis. Our results can be potentially applied to other provable learning algorithms as well [24, 31, 10, 30, 8], which might be of independent interest. We discuss our techniques and other related papers in the supplementary material.

## 2 Preliminary

Before proceeding, we define some notations and clarify our problem setup in this section.

**Notations:** We will use capital or lower-case letters to represent random variables, $\mathcal{D}$ to represent an $s$-concave distribution, and $\mathcal{D}_{u,t}$ to represent the conditional distribution of $\mathcal{D}$ over the set $\{x : |u \cdot x| \leq t\}$. We define the *sign* function as $\text{sign}(x) = +1$ if $x \geq 0$ and $-1$ otherwise. We denote by $B(\alpha, \beta) = \int_0^1 t^{\alpha-1}(1-t)^{\beta-1}dt$ the beta function, and $\Gamma(\alpha) = \int_0^\infty t^{\alpha-1}e^{-t}dt$ the gamma function. We will consider a single norm for the vectors in $\mathbb{R}^n$, namely, the 2-norm denoted by $\|x\|$. We will frequently use $\mu$ (or $\mu_f$, $\mu_{\mathcal{D}}$) to represent the measure of the probability distribution $\mathcal{D}$ with density function $f$. The notation $\text{ball}(w^*, t)$ represents the set $\{w \in \mathbb{R}^n : \|w - w^*\| \leq t\}$. For convenience, the symbol $\oplus$ slightly differs from the ordinary addition $+$: For $f = 0$ or $g = 0$, $\{f^s \oplus g^s\}^{1/s} = 0$; Otherwise, $\oplus$ and $+$ are the same. For $u, v \in \mathbb{R}^n$, we define the angle between them as $\theta(u, v)$.

## 2.1 From Log-Concavity to S-Concavity

We begin with the definition of $s$-concavity. There are slight differences among the definitions of $s$-concave density, $s$-concave distribution, and $s$-concave measure.

**Definition 1** (S-Concave (Density) Function, Distribution, Measure). *A function $f: \mathbb{R}^n \to \mathbb{R}_+$ is $s$-concave, for $-\infty \le s \le 1$, if $f(\lambda x + (1-\lambda)y) \ge (\lambda f(x)^s + (1-\lambda)f(y)^s)^{1/s}$ for all $\lambda \in [0,1]$, $\forall x, y \in \mathbb{R}^n$.[2] A probability distribution $\mathcal{D}$ is $s$-concave, if its density function is $s$-concave. A probability measure $\mu$ is $s$-concave if $\mu(\lambda A + (1-\lambda)B) \ge [\lambda \mu(A)^s + (1-\lambda)\mu(B)^s]^{1/s}$ for any sets $A, B \subseteq \mathbb{R}^n$ and $\lambda \in [0,1]$.*

Special classes of $s$-concave functions include *concavity* $(s = 1)$, *harmonic-concavity* $(s = -1)$, *quasi-concavity* $(s = -\infty)$, etc. The conditions in Definition 1 are progressively weaker as $s$ becomes smaller: $s_1$-concave densities (distributions, measures) are $s_2$-concave if $s_1 \ge s_2$. Thus one can verify [13]: concave $(s = 1) \subsetneq$ log-concave $(s = 0) \subsetneq s$-concave $(s < 0) \subsetneq$ quasi-concave $(s = -\infty)$.

# 3 Structural Results of S-Concave Distributions: A Toolkit

In this section, we develop geometric properties of $s$-concave distribution. The challenge is that unlike the commonly used distributions in learning (uniform or more generally log-concave distributions), this broader class is not closed under the marginalization operator and many such distributions are fat-tailed. To address this issue, we introduce several new techniques. We first introduce the extension of the Prekopa-Leindler inequality so as to reduce the high-dimensional problem to the one-dimensional case. We then reduce the resulting one-dimensional $s$-concave function to a well-defined baseline function, and explore the geometric properties of that baseline function. We summarize our high-level proof ideas briefly by the following figure.

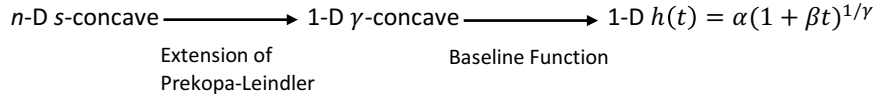

## 3.1 Marginal Distribution and Cumulative Distribution Function

We begin with the analysis of the marginal distribution, which forms the basis of other geometric properties of $s$-concave distributions $(s \le 0)$. Unlike the (nearly) log-concave distribution where the marginal remains (nearly) log-concave, the class of $s$-concave distributions is not closed under the marginalization operator. To study the marginal, our primary tool is the theory of convex geometry. Specifically, we will use an extension of the Prékopa-Leindler inequality developed by Brascamp and Lieb [11], which allows for a characterization of the integral of $s$-concave functions.

**Theorem 2** ([11], Thm 3.3). *Let $0 < \lambda < 1$, and $H_s$, $G_1$, and $G_2$ be non-negative integrable functions on $\mathbb{R}^m$ such that $H_s(\lambda x + (1-\lambda)y) \ge [\lambda G_1(x)^s \oplus (1-\lambda)G_2(y)^s]^{1/s}$ for every $x, y \in \mathbb{R}^m$. Then $\int_{\mathbb{R}^m} H_s(x)dx \ge \left[ \lambda \left( \int_{\mathbb{R}^m} G_1(x)dx \right)^\gamma + (1-\lambda) \left( \int_{\mathbb{R}^m} G_2(x)dx \right)^\gamma \right]^{1/\gamma}$ for $s \ge -1/m$, with $\gamma = s/(1+ms)$.*

Building on this, the following theorem plays a key role in our analysis of the marginal distribution.

**Theorem 3** (Marginal). *Let $f(x,y)$ be an $s$-concave density on a convex set $K \subseteq \mathbb{R}^{n+m}$ with $s \ge -\frac{1}{m}$. Denote by $K|_{\mathbb{R}^n} = \{x \in \mathbb{R}^n : \exists y \in \mathbb{R}^m \text{ s.t. } (x,y) \in K\}$. For every $x$ in $K|_{\mathbb{R}^n}$, consider the section $K(x) \triangleq \{y \in \mathbb{R}^m : (x,y) \in K\}$. Then the marginal density $g(x) \triangleq \int_{K(x)} f(x,y)dy$ is $\gamma$-concave on $K|_{\mathbb{R}^n}$, where $\gamma = \frac{s}{1+ms}$. Moreover, if $f(x,y)$ is isotropic, then $g(x)$ is isotropic.*

Similar to the marginal, the CDF of an $s$-concave distribution might not remain in the same class. This is in sharp contrast to log-concave distributions. The following theorem studies the CDF of an $s$-concave distribution.

**Theorem 4.** *The CDF of $s$-concave distribution in $\mathbb{R}^n$ is $\gamma$-concave, where $\gamma = \frac{s}{1+ns}$ and $s \ge -\frac{1}{n}$.*

Theorem 3 and 4 serve as the bridge that connects high-dimensional $s$-concave distributions to one-dimensional $\gamma$-concave distributions. With them, we are able to reduce the high-dimensional problem to the one-dimensional one.

## 3.2 Fat-Tailed Density

Tail probability is one of the most distinct characteristics of $s$-concave distributions compared to (nearly) log-concave distributions. While it can be shown that the (nearly) log-concave distribution has an exponentially small tail (Theorem 11, [7]), the tail of an $s$-concave distribution is fat, as proved by the following theorem.

**Theorem 5** (Tail Probability). *Let $x$ come from an isotropic distribution over $\mathbb{R}^n$ with an $s$-concave density. Then for every $t \geq 16$, we have $\Pr[\|x\| > \sqrt{n}t] \leq \left[1 - \frac{cst}{1+ns}\right]^{(1+ns)/s}$, where $c$ is an absolute constant.*

Theorem 5 is almost tight for $s < 0$. To see this, consider $X$ that is drawn from a one-dimensional Pareto distribution with density $f(x) = (-1 - \frac{1}{s})^{-\frac{1}{s}} x^{\frac{1}{s}}$ $(x \geq \frac{s+1}{-s})$. It can be easily seen that $\Pr[X > t] = \left[\frac{-s}{s+1}t\right]^{\frac{s+1}{s}}$ for $t \geq \frac{s+1}{-s}$, which matches Theorem 5 up to an absolute constant factor.

## 3.3 Geometry of S-Concave Distributions

We now investigate the geometry of $s$-concave distributions. We first consider one-dimensional $s$-concave distributions: We provide bounds on the density of centroid-centered halfspaces (Lemma 6) and range of the density function (Lemma 7). Building upon these, we develop geometric properties of high-dimensional $s$-concave distributions by reducing the distributions to the one-dimensional case based on marginalization (Theorem 3).

### 3.3.1 One-Dimensional Case

We begin with the analysis of one-dimensional halfspaces. To bound the probability, a normal technique is to bound the centroid region and the tail region separately. However, the challenge is that the $s$-concave distribution is fat-tailed (Theorem 5). So while the probability of a one-dimensional halfspace is bounded below by an absolute constant for log-concave distributions, such a probability for $s$-concave distributions decays as $s$ $(\leq 0)$ becomes smaller. The following lemma captures such an intuition.

**Lemma 6** (Density of Centroid-Centered Halfspaces). *Let $X$ be drawn from a one-dimensional distribution with $s$-concave density for $-1/2 \leq s \leq 0$. Then $\Pr(X \geq \mathbb{E}X) \geq (1+\gamma)^{-1/\gamma}$ for $\gamma = s/(1+s)$.*

We also study the image of a one-dimensional $s$-concave density. The following condition for $s > -1/3$ is for the existence of second-order moment.

**Lemma 7.** *Let $g : \mathbb{R} \to \mathbb{R}_+$ be an isotropic $s$-concave density function and $s > -1/3$. (a) For all $x$, $g(x) \leq \frac{1+s}{1+3s}$; (b) We have $g(0) \geq \sqrt{\frac{1}{3(1+\gamma)^{3/\gamma}}}$, where $\gamma = \frac{s}{s+1}$.*

### 3.3.2 High-Dimensional Case

We now move on to the high-dimensional case $(n \geq 2)$. In the following, we will assume $-\frac{1}{2n+3} \leq s \leq 0$. Though this working range of $s$ vanishes as $n$ becomes larger, it is almost the broadest range of $s$ that we can hopefully achieve: Chandrasekaran et al. [13] showed a lower bound of $s \geq -\frac{1}{n-1}$ if one require the $s$-concave distribution to have good geometric properties. In addition, we can see from Theorem 3 that if $s < -\frac{1}{n-1}$, the marginal of an $s$-concave distribution might even not exist; Such a case does happen for certain $s$-concave distributions with $s < -\frac{1}{n-1}$, e.g., the Cauchy distribution. So our range of $s$ is almost tight up to a $1/2$ factor.

We start our analysis with the density of centroid-centered halfspaces in high-dimensional spaces.

**Lemma 8** (Density of Centroid-Centered Halfspaces). *Let $f : \mathbb{R}^n \to \mathbb{R}_+$ be an $s$-concave density function, and let $H$ be any halfspace containing its centroid. Then $\int_H f(x)dx \geq (1+\gamma)^{-1/\gamma}$ for $\gamma = s/(1+ns)$.*

*Proof.* W.L.O.G., we assume $H$ is orthogonal to the first axis. By Theorem 3, the first marginal of $f$ is $s/(1+(n-1)s)$-concave. Then by Lemma 6, $\int_H f(x)dx \geq (1+\gamma)^{-1/\gamma}$, where $\gamma = s/(1+ns)$. $\square$

The following theorem is an extension of Lemma 7 to high-dimensional spaces. The proofs basically reduce the $n$-dimensional density to its first marginal by Theorem 3, and apply Lemma 7 to bound the image.

**Theorem 9** (Bounds on Density). *Let $f : \mathbb{R}^n \to \mathbb{R}_+$ be an isotropic $s$-concave density. Then*

*(a) Let $d(s,n) = (1+\gamma)^{-1/\gamma} \frac{1+3\beta}{3+3\beta}$, where $\beta = \frac{s}{1+(n-1)s}$ and $\gamma = \frac{\beta}{1+\beta}$. For any $u \in \mathbb{R}^n$ such that $\|u\| \leq d(s,n)$, we have $f(u) \geq \left( \frac{\|u\|}{d}((2 - 2^{-(n+1)s})^{-1} - 1) + 1 \right)^{1/s} f(0)$.*

*(b) $f(x) \leq f(0) \left[ \left( \frac{1+\beta}{1+3\beta} \sqrt{3(1+\gamma)^{3/\gamma}} 2^{n-1+1/s} \right)^s - 1 \right]^{1/s}$ for every $x$.*

*(c) There exists an $x \in \mathbb{R}^n$ such that $f(x) > (4e\pi)^{-n/2}$.*

*(d) $(4e\pi)^{-n/2} \left[ \left( \frac{1+\beta}{1+3\beta} \sqrt{3(1+\gamma)^{3/\gamma}} 2^{n-1+\frac{1}{s}} \right)^s - 1 \right]^{-\frac{1}{s}} < f(0) \leq (2 - 2^{-(n+1)s})^{1/s} \frac{n\Gamma(n/2)}{2\pi^{n/2}d^n}$.*

*(e) $f(x) \leq (2 - 2^{-(n+1)s})^{1/s} \frac{n\Gamma(n/2)}{2\pi^{n/2}d^n} \left[ \left( \frac{1+\beta}{1+3\beta} \sqrt{3(1+\gamma)^{3/\gamma}} 2^{n-1+1/s} \right)^s - 1 \right]^{1/s}$ for every $x$.*

*(f) For any line $\ell$ through the origin, $\int_\ell f \leq (2 - 2^{-ns})^{1/s} \frac{(n-1)\Gamma((n-1)/2)}{2\pi^{(n-1)/2}d^{n-1}}$.*

Theorem 9 provides uniform bounds on the density function. To obtain more refined upper bound on the image of $s$-concave densities, we have the following lemma. The proof is built upon Theorem 9.

**Lemma 10** (More Refined Upper Bound on Densities). *Let $f : \mathbb{R}^n \to \mathbb{R}_+$ be an isotropic $s$-concave density. Then $f(x) \leq \beta_1(n,s)(1 - s\beta_2(n,s)\|x\|)^{1/s}$ for every $x \in \mathbb{R}^n$, where*

$$\beta_1(n,s) = \frac{(2 - 2^{-(n+1)s})^{\frac{1}{s}}}{2\pi^{n/2}d^n}(1-s)^{-1/s}n\Gamma(n/2) \left[ \left( \frac{1+\beta}{1+3\beta} \sqrt{3(1+\gamma)^{3/\gamma}} 2^{n-1+1/s} \right)^s - 1 \right]^{1/s},$$

$$\beta_2(n,s) = \frac{2\pi^{(n-1)/2}d^{n-1}}{(n-1)\Gamma((n-1)/2)}(2 - 2^{-ns})^{-\frac{1}{s}} \frac{[(a(n,s) + (1-s)\beta_1(n,s)^s)^{1+\frac{1}{s}} - a(n,s)^{1+\frac{1}{s}}]s}{\beta_1(n,s)^s(1+s)(1-s)},$$

$$a(n,s) = (4e\pi)^{-ns/2} \left[ \left( \frac{1+\beta}{1+3\beta} \sqrt{3(1+\gamma)^{3/\gamma}} 2^{n-1+1/s} \right)^s - 1 \right]^{-1}, \gamma = \frac{\beta}{1+\beta}, \beta = \frac{s}{1+(n-1)s}, \text{ and}$$

$d = (1+\gamma)^{-\frac{1}{\gamma}} \frac{1+3\beta}{3+3\beta}$.

We also give an *absolute* bound on the measure of band.

**Theorem 11** (Probability inside Band). *Let $\mathcal{D}$ be an isotropic $s$-concave distribution in $\mathbb{R}^n$. Denote by $f_3(s,n) = 2(1+ns)/(1+(n+2)s)$. Then for any unit vector $w$, $\Pr_{x \sim \mathcal{D}}[|w \cdot x| \leq t] \leq f_3(s,n)t$. Moreover, if $t \leq d(s,n) \triangleq \left( \frac{1+2\gamma}{1+\gamma} \right)^{-\frac{1+\gamma}{\gamma}} \frac{1+3\gamma}{3+3\gamma}$ where $\gamma = \frac{s}{1+(n-1)s}$, then $\Pr_{x \sim \mathcal{D}}[|w \cdot x| \leq t] > f_2(s,n)t$, where $f_2(s,n) = 2(2 - 2^{-2\gamma})^{-1/\gamma}(4e\pi)^{-1/2} \left( 2 \left( \frac{1+\gamma}{1+3\gamma} \sqrt{3} \left( \frac{1+2\gamma}{1+\gamma} \right)^{\frac{3+3\gamma}{2\gamma}} \right)^\gamma - 1 \right)^{-1/\gamma}$.*

To analyze the problem of learning linear separators, we are interested in studying the disagreement between the hypothesis of the output and the hypothesis of the target. The following theorem captures such a characteristic under $s$-concave distributions.

**Theorem 12** (Probability of Disagreement). *Assume $\mathcal{D}$ is an isotropic $s$-concave distribution in $\mathbb{R}^n$. Then for any two unit vectors $u$ and $v$ in $\mathbb{R}^n$, we have $d_{\mathcal{D}}(u,v) = \Pr_{x \sim \mathcal{D}}[\text{sign}(u \cdot x) \neq \text{sign}(v \cdot x)] \geq f_1(s,n)\theta(u,v)$, where $f_1(s,n) = c(2 - 2^{-3\alpha})^{-\frac{1}{\alpha}} \left[ \left( \frac{1+\beta}{1+3\beta} \sqrt{3(1+\gamma)^{3/\gamma}} 2^{1+1/\alpha} \right)^\alpha - 1 \right]^{-\frac{1}{\alpha}} (1 + \gamma)^{-2/\gamma} \left( \frac{1+3\beta}{3+3\beta} \right)^2$, $c$ is an absolute constant, $\alpha = \frac{s}{1+(n-2)s}$, $\beta = \frac{s}{1+(n-1)s}$, $\gamma = \frac{s}{1+ns}$.*

Due to space constraints, all missing proofs are deferred to the supplementary material.

## 4   Applications: Provable Algorithms under S-Concave Distributions

In this section, we show that many algorithms that work under log-concave distributions behave well under $s$-concave distributions by applying the above-mentioned geometric properties. For simplicity, we will frequently use the notations in Theorem 1.

## 4.1 Margin Based Active Learning

We first investigate margin-based active learning under isotropic $s$-concave distributions in both *realizable* and *adversarial noise* models. The algorithm (see Algorithm 1) follows a localization technique: It proceeds in rounds, aiming to cut the error down by half in each round in the margin [6].

---

**Algorithm 1** Margin Based Active Learning under S-Concave Distributions

---

   **Input:** Parameters $b_k$, $\tau_k$, $r_k$, $m_k$, $\kappa$, and $T$ as in Theorem 16.
   **1:** Draw $m_1$ examples from $\mathcal{D}$, label them and put them into $W$.
   **2:** **For** $k = 1, 2, ..., T$
   **3:**    Find $v_k \in \mathsf{ball}(w_{k-1}, r_k)$ to approximately minimize the hinge loss over $W$ s.t. $\|v_k\| \leq 1$:
       $\ell_{\tau_k} \leq \min_{w \in \mathsf{ball}(w_{k-1}, r_k) \cap \mathsf{ball}(0,1)} \ell_{\tau_k}(w, W) + \kappa/8$.
   **4:**    Normalize $v_k$, yielding $w_k = \frac{v_k}{\|v_k\|}$; Clear the working set $W$.
   **5:**    **While** $m_{k+1}$ additional data points are not labeled
   **6:**       Draw sample $x$ from $\mathcal{D}$.
   **7:**       **If** $|w_k \cdot x| \geq b_k$, reject $x$; **else** ask for label of $x$ and put into $W$.
   **Output:** Hypothesis $w_T$.

---

### 4.1.1 Relevant Properties of S-Concave Distributions

The analysis requires more refined geometric properties as below. Theorem 13 basically claims that the error mostly concentrates in a band, and Theorem 14 guarantees that the variance in any 1-D direction cannot be too large. We defer the detailed proofs to the supplementary material.

**Theorem 13** (Disagreement outside Band). *Let $u$ and $v$ be two vectors in $\mathbb{R}^n$ and assume that $\theta(u, v) = \theta < \pi/2$. Let $\mathcal{D}$ be an isotropic $s$-concave distribution. Then for any absolute constant $c_1 > 0$ and any function $f_1(s, n) > 0$, there exists a function $f_4(s, n) > 0$ such that $\Pr_{x \sim \mathcal{D}}[\mathrm{sign}(u \cdot x) \neq \mathrm{sign}(v \cdot x)$ and $|v \cdot x| \geq f_4(s, n)\theta] \leq c_1 f_1(s, n)\theta$, where $f_4(s, n) = \frac{4\beta_1(2, \alpha)B(-1/\alpha - 3, 3)}{-c_1 f_1(s, n)\alpha^3 \beta_2(2, \alpha)^3}$, $B(\cdot, \cdot)$ is the beta function, $\alpha = s/(1 + (n - 2)s)$, $\beta_1(2, \alpha)$ and $\beta_2(2, \alpha)$ are given by Lemma 10.*

**Theorem 14** (1-D Variance). *Assume that $\mathcal{D}$ is isotropic $s$-concave. For $d$ given by Theorem 9 (a), there is an absolute $C_0$ such that for all $0 < t \leq d$ and for all $a$ such that $\|u - a\| \leq r$ and $\|a\| \leq 1$, $\mathbb{E}_{x \sim \mathcal{D}_{u,t}}[(a \cdot x)^2] \leq f_5(s, n)(r^2 + t^2)$, where $f_5(s, n) = 16 + C_0 \frac{8\beta_1(2, \eta)B(-1/\eta - 3, 2)}{f_2(s, n)\beta_2(2, \eta)^3 (\eta + 1)\eta^2}$, $(\beta_1(2, \eta), \beta_2(2, \eta))$ and $f_2(s, n)$ are given by Lemma 10 and Theorem 11, and $\eta = \frac{s}{1 + (n - 2)s}$.*

### 4.1.2 Realizable Case

We show that margin-based active learning works under $s$-concave distributions in the realizable case.

**Theorem 15.** *In the realizable case, let $\mathcal{D}$ be an isotropic $s$-concave distribution in $\mathbb{R}^n$. Then for $0 < \epsilon < 1/4$, $\delta > 0$, and absolute constants $c$, there is an algorithm (see the supplementary material) that runs in $T = \lceil \log \frac{1}{c\epsilon} \rceil$ iterations, requires $m_k = O\Big( \frac{f_3 \min\{2^{-k} f_4 f_1^{-1}, d\}}{2^{-k}} \Big( n \log \frac{f_3 \min\{2^{-k} f_4 f_1^{-1}, d\}}{2^{-k}} + \log \frac{1 + s - k}{\delta} \Big) \Big)$ labels in the k-th round, and outputs a linear separator of error at most $\epsilon$ with probability at least $1 - \delta$. In particular, when $s \to 0$ (a.k.a. log-concave), we have $m_k = O\big( n + \log(\frac{1 + s - k}{\delta}) \big)$.*

By Theorem 15, we see that the algorithm of margin-based active learning under $s$-concave distributions works almost as well as the log-concave distributions in the resizable case, improving exponentially w.r.t. the variable $1/\epsilon$ over passive learning algorithms.

### 4.1.3 Efficient Learning with Adversarial Noise

In the adversarial noise model, an adversary can choose any distribution $\widetilde{\mathcal{P}}$ over $\mathbb{R}^n \times \{+1, -1\}$ such that the marginal $\mathcal{D}$ over $\mathbb{R}^n$ is $s$-concave but an $\eta$ fraction of labels can be flipped adversarially. The analysis builds upon an induction technique where in each round we do hinge loss minimization in the band and cut down the 0/1 loss by half. The algorithm was previously analyzed in [3, 4] for the special class of log-concave distributions. In this paper, we analyze it for the much more general class of $s$-concave distributions.

**Theorem 16.** *Let $\mathcal{D}$ be an isotropic $s$-concave distribution in $\mathbb{R}^n$ over $x$ and the label $y$ obey the adversarial noise model. If the rate $\eta$ of adversarial noise satisfies $\eta < c_0 \epsilon$ for some absolute constant $c_0$, then for $0 < \epsilon < 1/4$, $\delta > 0$, and an absolute constant $c$, Algorithm 1 runs in $T = \lceil \log \frac{1}{c\epsilon} \rceil$ iterations, outputs a linear separator $w_T$ such that $\Pr_{x \sim \mathcal{D}}[\mathrm{sign}(w_T \cdot x) \neq \mathrm{sign}(w^* \cdot x)] \leq \epsilon$ with probability at least $1 - \delta$. The label complexity*

*in the k-th round is* $m_k = O\left(\frac{[b_{k-1}s+\tau_k(1+ns)[1-(\delta/(\sqrt{n}(k+k^2)))^{s/(1+ns)}]+\tau_k s]^2}{\kappa^2\tau_k^2 s^2}n\left(n+\log\frac{k+k^2}{\delta}\right)\right)$,
*where* $\kappa = \max\left\{\frac{f_3\tau_k}{f_2\min\{b_{k-1},d\}}, \frac{b_{k-1}\sqrt{f_5}}{\tau_k\sqrt{f_2}}\right\}$, $\tau_k = \Theta\left(f_1^{-2}f_2^{-1/2}f_3f_4^2f_5^{1/2}2^{-(k-1)}\right)$, *and* $b_k = \min\{\Theta(2^{-k}f_4f_1^{-1}),d\}$. *In particular, if* $s \to 0$, $m_k = O\left(n\log(\frac{n}{\epsilon\delta})(n+\log(\frac{k}{\delta}))\right)$.

By Theorem 16, the label complexity of margin-based active learning improves exponentially over that of passive learning w.r.t. $1/\epsilon$ even under fat-tailed $s$-concave distributions and challenging adversarial noise model.

## 4.2 Disagreement Based Active Learning

We apply our results to the analysis of disagreement-based active learning under $s$-concave distributions. The key is estimating the disagreement coefficient, a measure of complexity of active learning problems that can be used to bound the label complexity [20]. Recall the definition of the disagreement coefficient w.r.t. classifier $w^*$, precision $\epsilon$, and distribution $\mathcal{D}$ as follows. For any $r > 0$, define $\mathsf{ball}_{\mathcal{D}}(w,r) = \{u \in \mathcal{H} : d_{\mathcal{D}}(u,w) \leq r\}$ where $d_{\mathcal{D}}(u,w) = \Pr_{x\sim\mathcal{D}}[(u\cdot x)(w\cdot x) < 0]$. Define the disagreement region as $\mathrm{DIS}(\mathcal{H}) = \{x : \exists u,v \in \mathcal{H} \text{ s.t. } (u\cdot x)(v\cdot x) < 0\}$. Let the Alexander capacity $\mathsf{cap}_{w^*,\mathcal{D}} = \frac{\Pr_{\mathcal{D}}(\mathrm{DIS}(\mathsf{ball}_{\mathcal{D}}(w^*,r)))}{r}$. The disagreement coefficient is defined as $\Theta_{w^*,\mathcal{D}}(\epsilon) = \sup_{r\geq\epsilon}[\mathsf{cap}_{w^*,\mathcal{D}}(r)]$. Below, we state our results on the disagreement coefficient under isotropic $s$-concave distributions.

**Theorem 17** (Disagreement Coefficient). *Let $\mathcal{D}$ be an isotropic $s$-concave distribution over $\mathbb{R}^n$. For any $w^*$ and $r > 0$, the disagreement coefficient is* $\Theta_{w^*,\mathcal{D}}(\epsilon) = O\left(\frac{\sqrt{n}(1+ns)^2}{s(1+(n+2)s)f_1(s,n)}\left(1-\epsilon^{\frac{s}{1+ns}}\right)\right)$.
*In particular, when $s \to 0$ (a.k.a. log-concave), $\Theta_{w^*,\mathcal{D}}(\epsilon) = O(\sqrt{n}\log(1/\epsilon))$.*

Our bounds on the disagreement coefficient match the best known results for the much less general case of log-concave distributions [7]; Furthermore, they apply to the $s$-concave case where we allow arbitrary number of discontinuities, a case not captured by [18]. The result immediately implies concrete bounds on the label complexity of disagreement-based active learning algorithms, e.g., CAL [14] and $A^2$ [5]. For instance, by composing it with the result from [17], we obtain a bound of $\widetilde{O}\left(n^{3/2}\frac{(1+ns)^2}{s(1+(n+2)s)f(s)}(1-\epsilon^{s/(1+ns)})\left(\log^2\frac{1}{\epsilon}+\frac{OPT^2}{\epsilon^2}\right)\right)$ for *agnostic* active learning under an isotropic $s$-concave distribution $\mathcal{D}$. Namely, it suffices to output a halfspace with error at most $OPT + \epsilon$, where $OPT = \min_w \mathsf{err}_{\mathcal{D}}(w)$.

## 4.3 Learning Intersections of Halfspaces

Baum [9] provided a polynomial-time algorithm for learning the intersections of halfspaces w.r.t. symmetric distributions. Later, Klivans [27] extended the result by showing that the algorithm works under any distribution $\mathcal{D}$ as long as $\mu_{\mathcal{D}}(E) \approx \mu_{\mathcal{D}}(-E)$ for any set $E$. In this section, we show that it is possible to learn intersections of halfspaces under the broader class of s-concave distributions.

**Theorem 18.** *In the PAC realizable case, there is an algorithm (see the supplementary material) that outputs a hypothesis $h$ of error at most $\epsilon$ with probability at least $1-\delta$ under isotropic $s$-concave distributions. The label complexity is $M(\epsilon/2,\delta/4,n^2) + \max\{2m_2/\epsilon,(2/\epsilon^2)\log(4/\delta)\}$, where $M(\epsilon,\delta,m)$ is defined by $M(\epsilon,\delta,n) = O\left(\frac{n}{\epsilon}\log\frac{1}{\epsilon}+\frac{1}{\epsilon}\log\frac{1}{\delta}\right)$, $m_2 = M(\max\{\delta/(4eKm_1),\epsilon/2\},\delta/4,n)$, $K = \beta_1(3,\kappa)\frac{B(-1/\kappa-3,3)}{(-\kappa\beta_2(3,\kappa))^3}\frac{3+1/\kappa}{h(\kappa)d^{3+1/\kappa}}$, $d = (1+\gamma)^{-1/\gamma}\frac{1+3\beta}{3+3\beta}$,*
*$h(\kappa) = \left(\frac{1}{d}((2-2^{-4\kappa})^{-1}-1)+1\right)^{\frac{1}{\kappa}}(4e\pi)^{-\frac{3}{2}}\left[\left(\frac{1+\beta}{1+3\beta}\sqrt{3(1+\gamma)^{3/\gamma}}2^{2+\frac{1}{\kappa}}\right)^{\kappa}-1\right]^{-1/\kappa}$, $\beta = \frac{\kappa}{1+2\kappa}$,*
*$\gamma = \frac{\kappa}{1+\kappa}$, and $\kappa = \frac{s}{1+(n-3)s}$. In particular, if $s \to 0$ (a.k.a. log-concave), $K$ is an absolute constant.*

# 5 Lower Bounds

In this section, we give information-theoretic lower bounds on the label complexity of passive and active learning of homogeneous halfspaces under $s$-concave distributions.

**Theorem 19.** *For a fixed value $-\frac{1}{2n+3} \leq s \leq 0$ we have: (a) For any $s$-concave distribution $\mathcal{D}$ in $\mathbb{R}^n$ whose covariance matrix is of full rank, the sample complexity of learning origin-centered linear separators under $\mathcal{D}$ in the passive learning scenario is $\Omega(nf_1(s,n)/\epsilon)$; (b) The label complexity of active learning of linear separators under $s$-concave distributions is $\Omega(n\log(f_1(s,n)/\epsilon))$.*

If the covariance matrix of $\mathcal{D}$ is not of full rank, then the intrinsic dimension is less than $d$. So our lower bounds essentially apply to all $s$-concave distributions. According to Theorem 19, it is possible to have an exponential improvement of label complexity w.r.t. $1/\epsilon$ over passive learning by active sampling, even though the underlying distribution is a fat-tailed $s$-concave distribution. This observation is captured by Theorems 15 and 16.

# 6 Conclusions

In this paper, we study the geometric properties of $s$-concave distributions. Our work advances the state-of-the-art results on the margin-based active learning, disagreement-based active learning, and learning intersections of halfspaces w.r.t. the distributions over the instance space. When $s \to 0$, our results reduce to the best-known results for log-concave distributions. The geometric properties of $s$-concave distributions can be potentially applied to other learning algorithms, which might be of independent interest more broadly.

**Acknowledgements.** This work was supported in part by grants NSF-CCF 1535967, NSF CCF-1422910, NSF CCF-1451177, a Sloan Fellowship, and a Microsoft Research Fellowship.

## Footnotes

[2]When $s \to 0$, we note that $\lim_{s \to 0}(\lambda f(x)^s + (1-\lambda)f(y)^s)^{1/s} = \exp(\lambda \log f(x) + (1-\lambda)\log f(y))$. In this case, $f(x)$ is known to be log-concave.

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
