[Supplementary Material · supp.pdf]

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

# A Our Techniques

In this section, we introduce the techniques used for obtaining our results.

**Marginalization:** Our results are inspired by isoperimetric inequality for $s$-concave distributions by the work of Chandrasekaran et al. [23]. Roughly, the isoperimetry states that if two sets $K_1$ and $K_2$ are well-separated, then the area $B$ between them has large measure *relative to the measure of the two sets* (see Figure 1). Results of this kind are particularly useful for margin-based active learning of halfspace [5, 4, 6]: The algorithm proceeds in rounds, aiming to cut down the error by half in each round in the band. Since the measure of the band is large or even dominates, the error over the whole space decreases almost by half in each round, resulting in exponentially fast convergence rate. However, in order to make the analysis of such algorithms work for $s$-concave distribution, we typically require more refined geometric properties than the isoperimetry as the isoperimetric inequality states nothing about the *absolute* measure of band under $s$-concave distributions.

Figure 1: Isoperimetry.

The insight behind the isoperimetry is a collection of properties concerning the geometry of probability density. While the geometric properties of some classic paradigms, such as log-concave distributions (for the case of $s = 0$), are well-studied [49], it is typically hard to generalize those results to the $s$-concave distribution, for broader range of $s < 0$. This is due to the fact that the class of $s$-concave functions is not closed under marginalization: The marginal of an $s$-concave function may not be $s$-concave any more. This directly restricts the possibility of applying the prior proof techniques for log-concave distribution to the $s$-concave one. Furthermore, previous proofs heavily depend on the assumption that the density is light-tailed (see Theorem 11 in [9]), which is not applicable for probably fat-tailed $s$-concave distribution.

To mitigate the above concerns, we begin with a powerful tool from convex geometry by Brascamp and Lieb [20]. This result can be viewed as an extension of celebrated Prékopa-Leindler inequality, an integral inequality that is closely related to a number of classical inequalities in analysis and serves as the building block of isoperimetry under the log-concave distributions [21, 22]. With this, we can show that the marginal of any $s$-concave function is $\gamma$-concave, with a closed-form $\gamma$ that is related to the parameter $s$ and the dimension of marginalization. Our analysis is tight as there exists an $s$-concave function with a $\gamma$-concave marginal.

**Reduction to 1-D Baseline Function:** It is in general hard to study a high-dimensional $s$-concave distribution. Instead, we build on the marginalization technique described above to reduce each $n$-dimensional $s$-concave function to the one-dimensional case. Thus it suffices to investigate the geometry of one-dimensional $\gamma$-concave functions. But there are still infinitely many such functions in this class.

Our proofs take a novel analysis by reducing *all* one-dimensional $\gamma$-concave density to a certain baseline function. The baseline function should meet two goals: (a) It represents the worst case in the class of $\gamma$-concave functions, namely, such functions should achieve the bounds of geometric properties of our interest; (b) The function should be easy to analyze, e.g., with closed-form moments or integrations. Note that choosing a baseline function at the "boundary" between $\gamma$-concavity and non-$\gamma$-concavity classes readily achieves goal (a). To achieve goal (b), we set the "template" function as easy as $h(t) = \alpha(1 + \beta t)^{1/\gamma}$ for a particular choice of parameters $\alpha$ and $\beta$. Such functions have many good properties that one can exploit. First, the moments can be represented in closed-form by the beta function. This enables us to figure out the relations among moments of various orders explicitly and obtain a recursive inequality, which is critical for deducing the bounds of one-dimensional geometric properties. Second, $h(t)$ is at the "boundary" of $\gamma$-concave class: $h(t)^\eta$ is not a concave function for any $\eta < \gamma$. Therefore, this enables us to analyze the whole class of $s$-concavity by focusing on $h(t)$. Below, we summarize our high-level proof ideas briefly.

$$n\text{-D } s\text{-concave} \longrightarrow 1\text{-D } \gamma\text{-concave} \longrightarrow 1\text{-D } h(t) = \alpha(1 + \beta t)^{1/\gamma}$$

Extension of          Baseline Function
Prekopa-Leindler

# B    Additional Related Work

**Active Learning of Halfspace under Uniform Distribution:** Learning halfspace has been extensively studied in the past decades [16, 45, 29, 36, 52, 41, 40, 39]. Probably one of the most famous results is the VC argument. Vapnik [54] and Blumer et al. [17] showed that any hypothesis that is consistent with $\widetilde{O}(n/\epsilon)$ labeled examples has error at most $\epsilon$, if the VC dimension of the hypothesis class is $n$. The algorithm works under any data distribution and runs in polynomial time when the consistent hypothesis can be found efficiently, e.g., by linear programming in the realizable case. Other algorithms such as Perception [50], Winnow [47], and Support Vector Machine [55] provide better guarantees if the target vector has low $\ell_1$ or $\ell_2$ norm. All these results form the basis of passive learning.

To explore the possibility of further improving the label complexity, several algorithms were later proposed in the active learning literature [15, 14] under the uniform distributions [28, 30], among which disagreement-based active learning and margin-based active learning are two typical approaches. In the disagreement-based active learning, the algorithm proceeds in rounds, requesting the labels of instances in the disagreement region among the current candidate hypothesises. Cohn et al. [24] provided the first disagreement-based active learning algorithm in the realizable case. Balcan et al. [7] later extended such an algorithm to the agnostic setting by estimating the confidence interval of disagreement region. The analysis technique was further generalized thanks to Hanneke [34] by introducing the concept of disagreement coefficient, which is a new measure of complexity for active learning problems and serves as an important element for bounding the label complexity. However, this seminal work only focused on the disagreement coefficient under the uniform distribution.

Margin-based active learning is another line of research in the active learning literature. The algorithm proceeds in rounds, requesting labels of examples aggressively in the margin area around the current guess of hypothesis. Balcan et al. [8] first proposed an algorithm for margin-based active learning under the uniform distribution in the realizable case. They also provided guarantees under the *Tsybakov noise* model [53], but the algorithm is inefficient. To mitigate the issue, Awasthi et al. [3] considered a subclass of Tsybakov noise — *Massart noise* [19]. The algorithm runs in polynomial time by doing a sequence of hinge loss minimizations on the labeled instances. However, it was not clear then whether the analysis works for other distributions instead of the uniform one.

**Geometry of Log-Concave Distribution:** Log-concave distribution, a class of probability distributions such that the logarithm of density function is concave, is a common generalization of uniform distribution over the convex set [49]. Bertsimas and Vempala [12] and Kalai and Vempala [37] noticed that efficient sampling, integration, and optimization algorithms for this distribution class rely heavily on the good isoperimetry of density functions. Informally, a function has good isoperimetry if one cannot remove a small-measure set from its domain and partition the domain into two disjoint large-measure sets. The isoperimetry is commonly believed as a characteristic of good geometric properties. To see this, Lovász and Vempala [49] proved the isoperimetric inequality for the log-concave distribution, and provided a bunch of refined geometric properties for this distribution class. Going slightly beyond the log-concave distribution, Caramanis and Mannor [22] showed good isoperimetry for *nearly log-concave* distributions, but more refined geometry was not provided there.

Active learning of halfspace under (nearly) log-concave distribution has a natural connection to the geometry of that distribution (a.k.a. admissible distribution). The connection was first introduced by [9], and is sufficient for the success of disagreement-based and margin-based active learning under log-concave distribution [9]. To resolve the computational issue, Awasthi et al. [5] studied the probability of disagreement outside the margin under the log-concave distribution, and proposed an efficient algorithm for the challenging adversarial noise. More recently, Awasthi et al. [4] provided stronger guarantees for efficient learning of halfspace in the Massart noise model under log-concave distribution.

**S-Concave Distribution:** The problem of extending the log-concave distribution to the broader one for provable learning algorithms has received significant attention in recent years. Although some efforts have been devoted to generalizing the probability distribution, e.g., to the nearly log-concave distribution [9], the analysis is intrinsically built upon the geometry of log-concave distribution. Moreover, to the best of our knowledge, there is no *efficient, noise-tolerant* active learning algorithm that goes beyond the log-concave distribution. As a candidate extension, the class of $s$-concave distributions has many appealing properties that one can exploit [23, 33]: (a) The distribution class is much broader than the log-concave distributions as $s = 0$ implies the log-concavity; (b) The $s$-concave function mapping from $\mathbb{R}^n$ to $\mathbb{R}_+$ has good isoperimetry if $s \geq -1/(n-1)$; (c) Efficient sampling, integration, and optimization algorithms are available for such distribution class. All these properties inspire our work.

## C   Proof of Theorem 3

**Theorem 3 (restated)** *Let $f(x,y)$ be an $s$-concave density on a convex set $K \subseteq \mathbb{R}^{n+m}$ with $s \geq -\frac{1}{m}$. Denote by $K|_{\mathbb{R}^n} = \{x \in \mathbb{R}^n : \exists y \in \mathbb{R}^m \text{ s.t. } (x,y) \in K\}$. For every $x$ in $K|_{\mathbb{R}^n}$, consider the section $K(x) \triangleq \{y \in \mathbb{R}^m : (x,y) \in K\}$. Then the marginal density $g(x) \triangleq \int_{K(x)} f(x,y)dy$ is $\gamma$-concave on $K|_{\mathbb{R}^n}$, where $\gamma = \frac{s}{1+ms}$. Moreover, if $f(x,y)$ is isotropic, then $g(x)$ is isotropic.*

*Proof.* The proof that $g(x)$ is isotropic is standard [49]. We now prove the first part. Let $x_1$, $x_2$ be any two points. Define $g_i(y) = f(x_i, y)$ for $i = 1, 2$. So the functions $g_i(y)$ is defined on $K(x_i)$, $i = 1, 2$. Now let $x = \lambda x_1 + (1-\lambda)x_2$ for $\lambda \in (0,1)$ and define $h_s(y) = f(x,y)$ on $K(x)$. Notice that for any $y_i \in K(x_i)$, $i = 1, 2$, $y = \lambda y_1 + (1-\lambda)y_2 \in K(x)$. To see this, by the convexity of the set $K$, the point $(x, y) = \lambda(x_1, y_1) + (1-\lambda)(x_2, y_2)$ belongs to $K$. So $y \in K(x)$, i.e., $\lambda K(x_1) + (1-\lambda)K(x_2) \subseteq K(x)$. Using the $s$-concavity of $f(x,y)$, we have $f(x,y) = f(\lambda(x_1, y_1) + (1-\lambda)(x_2, y_2)) \geq [\lambda f(x_1, y_1)^s + (1-\lambda)f(x_2, y_2)^s]^{1/s}$, which implies that $h_s(y) = h_s(\lambda y_1 + (1-\lambda)y_2) \geq [\lambda g_1(y_1)^s + (1-\lambda)g_2(y_2)^s]^{1/s}$. Denote by $I_{(\cdot)}$ the indicator function. So $h_s(\lambda y_1 + (1-\lambda)y_2)I_{K(x)}(y) \geq \left[\lambda(g_1(y_1)I_{K(x_1)}(y_1))^s \oplus (1-\lambda)(g_2(y_2)I_{K(x_2)}(y_2))^s\right]^{1/s}$. Set $H_s(y) = h_s(\lambda y_1 + (1-\lambda)y_2)I_{K(x)}$, $G_1(y_1) = g_1(y_1)I_{K(x_1)}$, $G_2(y_1) = g_2(y_1)I_{K(x_2)}$. By Theorem 2,

$$g(x) = \int_{\mathbb{R}^m} H_s(y)dy = \int_{\mathbb{R}^m} h_s(y)I_{K(x)}(y)dy \geq \left[(1-\lambda)\left(\int_{\mathbb{R}^n} G_1(y)dy\right)^\gamma + \lambda\left(\int_{\mathbb{R}^n} G_2(y)dy\right)^\gamma\right]^{1/\gamma}$$

$$= \left[(1-\lambda)\left(\int_{\mathbb{R}^n} f(x_1, y_1)I_{K(x_1)}(y_1)dy_1\right)^\gamma + \lambda\left(\int_{\mathbb{R}^n} f(x_2, y_2)I_{K(x_2)}(y_2)dy_2\right)^\gamma\right]^{1/\gamma}$$

$$= [(1-\lambda)g(x_1)^\gamma + \lambda g(x_2)^\gamma]^{1/\gamma},$$

where $\gamma = s/(1+ms)$. Namely, the marginal function $g(x)$ is $\gamma$-concave. $\qquad\square$

## D   Proof of Theorem 4

Similar to the marginal, the CDF of an $s$-concave distribution might not remain in the same class. This is in sharp contrast with the log-concave distributions. The following lemma from [20] provides a useful tool for our analysis of CDF, which basically claims that the measure of any $s$-concave distribution is $\gamma$-concave with a closed-form $\gamma$.

**Lemma 20** ([20], Cor 3.4)**.** *The density function $f(x)$ is $s$-concave for $s \geq -1/n$ where $x \in \mathbb{R}^n$, if and only if the corresponding probability measure $\mu$ is $\gamma$-concave for $\gamma = \frac{s}{1+ns}$, namely, $\mu(\lambda A + (1-\lambda)B) \geq [\lambda\mu(A)^\gamma + (1-\lambda)\mu(B)^\gamma]^{1/\gamma}$, for any $A, B \subseteq \mathbb{R}^n$ and $\lambda \in [0,1]$, where $\mu(A) = \int_A f(x)dx$.*

Lemma 20 is an extension of celebrated Brunn-Minkowski theorem. The following theorem concerning the CDF of an $s$-concave distribution is a straightforward result from Lemma 20.

**Theorem 4 (restated)** *The CDF of $s$-concave distribution in $\mathbb{R}^n$ is $\gamma$-concave, where $\gamma = \frac{s}{1+ns}$ and $s \geq -\frac{1}{n}$.*

*Proof.* Denote by $F(x)$ the CDF. Applying Lemma 20 to the set $A = \{x : x \leq x_1\}$ and $B = \{x : x \leq x_2\}$ and taking into account that $F(x_1) = \mu(A)$, $F(x_2) = \mu(B)$, and $F(\lambda x_1 + (1 - \lambda)x_2) = \mu(\lambda A + (1 - \lambda)B)$, we have the result. $\square$

## E  Proof of Theorem 5

Tail probability is one of the most distinct characteristic of $s$-concave distributions compared to the nearly log-concavity. To study this, we first require a concentration result from [18].

**Lemma 21** ([18], Thm 5.2). *Let $f$ be a Borel function on $\mathbb{R}^n$ and let $m$ be a median for $|f|$ w.r.t. a $\kappa$-concave measure $\mu$, where $\kappa < 0$. Then for every $t > 1$ such that $4\delta_f(\frac{1}{t}) \leq 1$, we have $\mu[|f| > mt] \leq \left[1 - \frac{c\kappa}{\delta_f(\frac{1}{t})}\right]^{1/\kappa}$, where $c$ is a constant, $\delta_f(\epsilon) = \sup_{x,y} mes\{t \in (0,1) : |f(tx + (1-t)y)| \leq \epsilon|f(x)|\}$, $0 \leq \epsilon \leq 1$, and mes stands for the Lebesgue measure.*

Now we are ready to bound the tail probability of $s$-concave density. While it can be shown that the nearly log-concave distribution has an exponentially small tail (Theorem 11, [9]), the tail of $s$-concave distribution is fat, as proved by the following theorem.

**Theorem 5 (restated)** *Let $x$ come from an isotropic distribution over $\mathbb{R}^n$ with an $s$-concave density. Then for every $t \geq 16$, we have $\Pr[\|x\| > \sqrt{n}t] \leq \left[1 - \frac{cst}{1+ns}\right]^{(1+ns)/s}$, where $c$ is an absolute constant.*

*Proof.* Set function $f(x)$ in Lemma 21 as $\|x\|$. Bobkov [18] claimed that $\delta_f(\epsilon) \leq 2\epsilon$. Also, Lemma 20 implies that the probability measure is $\kappa = \frac{s}{1+ns}$-concave.

By the definition of the median $m$, the Markov inequality, and the Jensen inequality, we have $\frac{1}{2} = \Pr[\|x\| \geq m] \leq \frac{\mathbb{E}\|x\|}{m} \leq \frac{\sqrt{\mathbb{E}\|x\|^2}}{m} = \frac{\sqrt{n}}{m}$, where the last equality is due to the isotropicity assumption. So by Lemma 21, we have that for every $t \geq 8$, $\Pr[\|x\| > 2\sqrt{n}t] \leq \Pr[\|x\| > mt] \leq [1 - cst/(1+ns)]^{(1+ns)/s}$. Replacing $t$ with $t/2$, the proof is completed. $\square$

## F  Proof of Lemma 6

**Lemma 6 (restated)** *Let $X$ be a random variable drawn from a one-dimensional distribution with $s$-concave density for $-1/2 \leq s \leq 0$. Then*

$$\Pr(X \geq \mathbb{E}X) \geq (1 + \gamma)^{-1/\gamma},$$

*for $\gamma = s/(1+s)$.*

*Proof.* Without loss of generality, we assume that $\mathbb{E}X = 0$ and $|X| \leq K$. The general case then follows by translation transformation and approximating a general distribution with $s$-concave density by such bounded distributions.

Let $G(x) = \Pr(X \leq x)$ be the CDF of the $s$-concave density. We first prove that $\Pr(X \leq \mathbb{E}X) \geq (1 + \gamma)^{-1/\gamma}$. By Theorem 4, $G(x)$ is $\gamma$-concave, monotone increasing such that $G(x) = 0$ for $x \leq -K$ and $G(x) = 1$ for $x \geq K$, where $-1 \leq \gamma = \frac{s}{1+s} \leq 0$. Notice that the assumption of centroid 0 implies that $\int_{-K}^{K} xG'(x)dx = 0$, which equivalently means $\int_{-K}^{K} G(x)dx = K$ by integration by parts. Our goal is to prove that $G(0) \geq (1 + \gamma)^{-1/\gamma}$.

The function $G^\gamma$ is concave for $\gamma < 0$. Thus it lies above its tangent at 0. This means that $G(x) \leq G(0)(1 + \gamma cx)^{\frac{1}{\gamma}}$, where $c = G'(0)/G(0) > 0$. We now set $K$ large enough so that $1/c < K$. Then

$$G(x) \leq \begin{cases} G(0)(1 + \gamma cx)^{\frac{1}{\gamma}}, & \text{if } x \leq 1/c, \\ 1, & \text{if } x > 1/c. \end{cases}$$

So

$$K = \int_{-K}^{K} G(x)dx$$

$$\leq \int_{-K}^{1/c} G(0)(1 + \gamma cx)^{\frac{1}{\gamma}} dx + \int_{1/c}^{K} 1 dx$$

$$= \frac{G(0)}{c(\gamma + 1)}[(1 + \gamma)^{\frac{1}{\gamma}+1} - (1 - \gamma cK)^{\frac{1}{\gamma}+1}] + K - \frac{1}{c}$$

$$\leq \frac{G(0)(1 + \gamma)^{\frac{1}{\gamma}}}{c} + K - \frac{1}{c},$$

which implies that $G(0) \geq (1 + \gamma)^{-1/\gamma}$ as claimed. Replacing $X$ with $Y = -X$, we obtain the result. $\square$

## G   Proof of Lemma 7

As a preliminary, we first prove the following lemma concerning the moments of $s$-concave distribution.

**Lemma 22.** *Let $g : \mathbb{R}_+ \to \mathbb{R}_+$ be an integrable function. Define $M_n(g) = \int_0^\infty t^n g(t)dt$, and suppose it exists. Then*

*(a) The sequence $\{M_n(g) : n = 0, 1, ...\}$ is log-convex, which means $\log M_n(g)$ is convex w.r.t. variable $n$, or equivalently $M_n(g)M_{n+2}(g) \geq M_{n+1}(g)^2$ for any $n \in N$.*

*(b) If $g$ is monotone decreasing, then the sequence defined by*

$$M'_n(g) = \begin{cases} nM_{n-1}(g), & \text{if } n > 0, \\ g(0), & \text{if } n = 0, \end{cases}$$

*is log-convex.*

*(c) If $g$ is $s$-concave ($s > -1/(n+1)$), then the sequence $T_n(g) \triangleq M_n(g)/B(-1/s-n-1, n+1)$ is log-concave, which means $\log T_n(g)$ is concave w.r.t. $n$, or equivalently $T_n(g)T_{n+2}(g) \leq T_{n+1}(g)^2$ for any $n \in N$.*

*(d) If $g$ is $s$-concave, then $g(0)M_1(g) \leq M_0(g)^2 \frac{1+s}{1+2s}$.*

*Proof.* The proofs of Parts (a) and (b) are from [49].

(c) The intuition behind the proof is to choose a baseline $s$-concave function $h$ which is at the "boundary" between the family of $s$-concave function and that of the non $s$-concave function. We show that $h$ satisfies the equation

$$T_n(h)T_{n+2}(h) = T_{n+1}^2(h). \tag{1}$$

Then by the facts that $h$ is at the "boundary" and $g$ is any $s$-concave function, we have

$$T_{n+1}(h) \leq T_{n+1}(g). \tag{2}$$

The conclusion follows from (1) and (2), and from our choice of $h$ such that $T_n(h) = T_n(g)$ and $T_{n+2}(h) = T_{n+2}(g)$, by adjusting the slope and intercept of the linear function.

Formally, let $h(t) = \beta(1 + \gamma t)^{1/s}$ be the above-mentioned baseline $s$-concave function ($\beta, \gamma > 0$) such that

$$M_n(h) = M_n(g) \quad \text{and} \quad M_{n+2}(h) = M_{n+2}(g)$$

(This holds because there are two parameters $\beta, \gamma$ and two equations). That means

$$\int_0^\infty t^n(h(t) - g(t))dt = 0 \quad \text{and} \quad \int_0^\infty t^{n+2}(h(t) - g(t))dt = 0.$$

Then it follows that the graph of $h$ must intersect the graph of $g$ at least twice. Since $g$ is $s$-concave, which implies the uni-modality, the graphs of $h$ and $g$ intersect exactly at two points $0 \leq a < b$.

Moreover, $h \leq g$ in the interval $[a, b]$ and $h \geq g$ outside the interval. That is to say, $(t-a)(t-b)$ has the same sign as $h - g$. Thus

$$\int_0^\infty (t-a)(t-b)t^n(h(t) - g(t))dt \geq 0.$$

Namely,

$$0 = \int_0^\infty t^{n+2}(h(t) - g(t))dt + ab \int_0^\infty t^n(h(t) - g(t))dt \geq (a+b) \int_0^\infty t^{n+1}(h(t) - g(t))dt.$$

This implies that

$$M_{n+1}(h) = \int_0^\infty t^{n+1}h(t)dt \leq \int_0^\infty t^{n+1}g(t)dt = M_{n+1}(g).$$

Since

$$M_n(h) = \int_0^\infty t^n \beta(1+\gamma t)^{1/s}dt = B(-1/s - n - 1, n+1)\frac{\beta}{\gamma^{n+1}}$$

for $s > -1/(n+1)$, we have

$$\frac{M_n(g)}{B(-1/s - n - 1, n+1)} \frac{M_{n+2}(g)}{B(-1/s - n - 3, n+3)} = \frac{M_n(h)}{B(-1/s - n - 1, n+1)} \frac{M_{n+2}(h)}{B(-1/s - n - 3, n+3)}$$

$$= \frac{\beta}{\gamma^{n+1}} \cdot \frac{\beta}{\gamma^{n+3}}$$

$$= \left(\frac{M_{n+1}(h)}{B(-1/s - n - 2, n+2)}\right)^2$$

$$\leq \left(\frac{M_{n+1}(g)}{B(-1/s - n - 2, n+2)}\right)^2,$$

as desired.

(d) The proof is almost the same as that of Part (c). Let $h(t) = \beta(1+\gamma t)^{1/s}$ be an $s$-concave function $(\beta, \gamma > 0)$ such that

$$h(0) = g(0) \quad \text{and} \quad M_1(h) = M_1(g).$$

So the graphs of $h$ and $g$ intersect exactly at two points $0$ and $a > 0$, and hence

$$\int_0^\infty t(t-a)t^{-1}(h(t) - g(t))dt \geq 0.$$

That means

$$0 = \int_0^\infty t(h(t) - g(t))dt \geq a \int_0^\infty (h(t) - g(t))dt,$$

or equivalently,

$$M_0(h) \leq M_0(g).$$

Note that $h(0)M_1(h) = M_0(h)^2 \frac{1+s}{1+2s}$ by (G). Then the conclusion follows by the fact

$$g(0)M_1(g) = h(0)M_1(h) = M_0(h)^2 \frac{1+s}{1+2s} \leq M_0(g)^2 \frac{1+s}{1+2s}.$$

$\square$

Now we are ready to prove Lemma 7.

**Lemma 7** (restated) *Let $g : \mathbb{R} \to \mathbb{R}_+$ be an isotropic $s$-concave density function and $s > -1/3$.*

*(a) For all $x$, $g(x) \leq \frac{1+s}{1+3s}$.*

*(b) We have $g(0) \geq \sqrt{\frac{1}{3(1+\gamma)^{3/\gamma}}}$, where $\gamma = \frac{s}{s+1}$.*

*Proof.* (a) Let $z$ be the maximum point of function $g$. Intuitively, if the value of the function $g$ evaluated at $z$ is too large, the corresponding distribution has a small deviation from $z$ (Second part of the proof below). However, the moment property of Lemma 22 restricts that the second moment cannot be too small (First part of the proof below), which leads to a contradiction.

Formally, suppose that $g(z) > \frac{1+s}{1+3s}$. Define

$$M_i = \int_z^\infty (x - z)^i g(x) dx, \quad \text{and} \quad N_i = \int_{-\infty}^z (z - x)^i g(x) dx.$$

By the isotropicity of function $g$, we have

$$M_0 + N_0 = 1, \quad N_1 - M_1 = z, \quad M_2 + N_2 = 1 + z^2.$$

Thus

$$\begin{aligned}
M_2 + N_2 &= (M_0 + N_0)^2 + (M_1 - N_1)^2 \\
&= (M_0 - M_1)^2 + (N_0 - N_1)^2 + 2(M_0 N_0 - M_1 N_1) + 2(M_0 M_1 + N_0 N_1) \\
&\geq 2(M_0 M_1 + N_0 N_1),
\end{aligned}$$

where the last inequality holds since, by Lemma 22 (d), we have $M_1 \leq \frac{M_0^2}{g(z)} \frac{1+s}{1+2s} \leq M_0^2 \leq M_0$ and $N_1 \leq \frac{N_0^2}{g(z)} \frac{1+s}{1+2s} \leq N_0^2 \leq N_0$.

On the other hand, by Lemma 22 (c) (d),

$$M_2 \leq \frac{2M_1^2}{M_0} \frac{1+2s}{1+3s} \leq \frac{2M_1 M_0}{g(z)} \frac{1+s}{1+3s} < 2M_1 M_0,$$

and similarly, $N_2 < 2N_1 N_0$. That means

$$M_2 + N_2 < 2(M_0 M_1 + N_0 N_1),$$

and we obtain a contradiction.

(b) The proof is by Lemma 22 (b) which lower bounds $g(0)$ by the second order moment of $g$, which is 1 according to isotropicity.

Specifically, without lose of generality, assume that $g(x)$ is monotone decreasing for $x \geq 0$ (otherwise consider $g(-x)$, since function $g$ is uni-modal). Define $g_0$ as the restriction of $g$ to the non-negative semi-line. Then by Lemma 22 (b), we have

$$M_1'(g_0)^3 \leq M_0'(g_0)^2 M_3'(g_0),$$

which implies

$$g(0) \geq \sqrt{\frac{M_0(g_0)^3}{3M_2(g_0)}}.$$

Note that $M_2(g_0) \leq M_2(g) = 1$, and by Lemma 6,

$$M_0(g_0) = \int_0^\infty g(t) dt = \Pr[X \geq \mathbb{E} X] \geq (1 + \gamma)^{-1/\gamma}.$$

Thus we have

$$g(0) \geq \sqrt{\frac{1}{3(1+\gamma)^{3/\gamma}}},$$

where $\gamma = s/(1+s)$. $\qquad\qquad\square$

## H    Proof of Theorem 9

**Theorem 9** (restated) *Let $f : \mathbb{R}^n \to \mathbb{R}_+$ be an isotropic s-concave density.*

*(a) Let $d = (1 + \gamma)^{-1/\gamma} \frac{1+3\beta}{3+3\beta}$, where $\beta = \frac{s}{1+(n-1)s}$ and $\gamma = \frac{\beta}{1+\beta}$. For any $u \in \mathbb{R}^n$ such that $\|u\| \leq d$, we have $f(u) \geq \left( \frac{\|u\|}{d}((2 - 2^{-(n+1)s})^{-1} - 1) + 1 \right)^{1/s} f(0)$.*

(b) $f(x) \leq f(0) \left[ \left( \frac{1+\beta}{1+3\beta} \sqrt{3(1+\gamma)^{3/\gamma}} 2^{n-1+1/s} \right)^s - 1 \right]^{1/s}$ *for every* $x$ $(s \geq -\frac{1}{2n+3})$.

(c) *There exists an* $x \in \mathbb{R}^n$ *such that* $f(x) > (4e\pi)^{-n/2}$.

(d) *We have* $(4e\pi)^{-n/2} \left[ \left( \frac{1+\beta}{1+3\beta} \sqrt{3(1+\gamma)^{3/\gamma}} 2^{n-1+1/s} \right)^s - 1 \right]^{-1/s}$ $<$ $f(0)$ $\leq$ $(2 - 2^{-(n+1)s})^{1/s} \frac{n\Gamma(n/2)}{2\pi^{n/2}d^n}$.

(e) $f(x) \leq (2 - 2^{-(n+1)s})^{1/s} \frac{n\Gamma(n/2)}{2\pi^{n/2}d^n} \left[ \left( \frac{1+\beta}{1+3\beta} \sqrt{3(1+\gamma)^{3/\gamma}} 2^{n-1+1/s} \right)^s - 1 \right]^{1/s}$ *for every* $x$.

(f) *For any line* $\ell$ *through the origin,* $\int_\ell f \leq (2 - 2^{-ns})^{1/s} \frac{(n-1)\Gamma((n-1)/2)}{2\pi^{(n-1)/2}d^{n-1}}$.

*Proof.* (a) Formally, suppose that the conclusion does not hold true, i.e., there is a point $u$ such that $\|u\| = t \leq d$ and $f(u) < \left( \frac{t}{d}((2 - 2^{-(n+1)s})^{-1} - 1) + 1 \right)^{1/s} f(0)$. Define $v = \frac{d}{t}u$ and note that $0 \leq \frac{t}{d} \leq 1$. Therefore, by the $s$-concavity of $f$, we have

$$f(u) = f\left( \frac{t}{d}v + \left(1 - \frac{t}{d}\right)0 \right) \geq \left[ \frac{t}{d}f(v)^s + \left(1 - \frac{t}{d}\right)f(0)^s \right]^{1/s},$$

which together with $f(u) < \left( \frac{t}{d}((2 - 2^{-(n+1)s})^{-1} - 1) + 1 \right)^{1/s} f(0)$ implies $f(v) < (2 - 2^{-(n+1)s})^{-1/s}f(0)$. Let $H$ be a hyperplane supporting the convex set $\{x \in \mathbb{R}^n : f(x) \geq f(v)\}$ through the point $v$ (the convexity follows from the $s$-concavity of $f$). Define an orthogonal coordinate system in which the hyperplane is parallel to coordinate plane so that it can be represented as $x_1 = a$ for some $0 < a \leq d$. Thus $f(x) < (2 - 2^{-(n+1)s})^{-1/s}f(0)$ for any $x$ such that $x_1 \geq a$. We will prove that this implies that the 1-dimensional marginal is not flat.

Denote by $g$ the first marginal of the $n$-dimensional function $f$. Then $g$ is isotropic and $\beta = \frac{s}{1+(n-1)s}$-concave by Theorem 3, and $g(x) \leq \frac{1+\beta}{1+3\beta}$ for all $x$ by Lemma 7 (a). We prove that

$$g(2b) < \frac{g(b)}{4}$$

for any $b \geq a$, which means that the 1-dimensional function is not flat. To see this, by the $s$-concavity of function $f$, we have that, for every $x$ such that $x_1 \geq a$,

$$f(2x)^s \geq 2f(x)^s - f(0)^s > 2^{-(n+1)s}f(x)^s.$$

Namely, $f(2x) < 2^{-(n+1)}f(x)$. Hence

$$g(2b) = \int_{(x_1=2b)} f(x)dx_2...dx_n < 2^{-(n+1)}2^{n-1} \int_{(x_1=b)} f(x)dx_2...dx_n = \frac{g(b)}{4}.$$

So

$$\int_a^\infty g(y)dy = \int_a^{2a} g(y)dy + \int_{2a}^\infty g(y)dy < \int_a^{2a} g(y)dy + \frac{1}{2} \int_a^\infty g(y)dy.$$

Namely, by Lemma 7 (a),

$$\int_a^\infty g(y)dy < 2 \int_a^{2a} g(y)dy \leq 2a \frac{1+\beta}{1+3\beta}.$$

So

$$\int_0^\infty g(y)dy = \int_0^a g(y)dy + \int_a^\infty g(y)dy < 3a\frac{1+\beta}{1+3\beta} \leq 3d\frac{1+\beta}{1+3\beta} = (1+\gamma)^{-1/\gamma},$$

which leads to a contradiction with Lemma 6.

(b) If $f(w) \leq f(0)$ for every $w$, then the conclusion holds true. Otherwise, let $w$ be the point such that $f(w) > f(0)$. Let $H_0$ be the hyperplane through $0$ which supports the convex set $\{x \in \mathbb{R}^n : f(x) \geq f(0)\}$. By defining an orthogonal system, we may set $H_0$ as the hyperplane

$x_1 = 0$, and so $f(x) \leq f(0)$ for any $x$ such that $x_1 = 0$. Define $g$, which is a $\beta = \frac{s}{1+(n-1)s}$-concave function, as the first marginal of function $f$. Denote by $H_t$ the hyperplane $x_1 = t$. Without loss of generality, we assume that $w \in H_b$ with $b > 0$.

Let $x$ be any point on $H_0$ and $x'$ be the intersection between line segment $[x, w]$ and $H_{b/2}$. Then by the $s$-concavity of $f$ and $f(x) \leq f(0)$ for $x \in H_0$, we have

$$f(x') \geq \left[ \frac{1}{2} f(x)^s + \frac{1}{2} f(w)^s \right]^{1/s} \geq \left( \frac{1}{2} \right)^{1/s} f(x) \left[ 1 + \left( \frac{f(w)}{f(0)} \right)^s \right]^{1/s}.$$

Thus

$$g(b/2) = \int_{(x_1=b/2)} f(x) dx_2 ... dx_n \geq \frac{1}{2^{n-1+1/s}} \left[ 1 + \left( \frac{f(w)}{f(0)} \right)^s \right]^{1/s} g(0).$$

By Lemma 7 (a) (b), we have

$$\frac{1+\beta}{1+3\beta} \geq g(b/2) \geq \frac{1}{2^{n-1+1/s}} \left[ 1 + \left( \frac{f(w)}{f(0)} \right)^s \right]^{1/s} \sqrt{\frac{1}{3(1+\gamma)^{3/\gamma}}},$$

where $\gamma = \beta/(1+\beta)$. Note that $s \geq -\frac{1}{2n+3}$ implies $\frac{1+\beta}{1+3\beta} 2^{n-1+1/s} \sqrt{3(1+\gamma)^{3/\gamma}} < 1$. So

$$f(w) \leq f(0) \left[ \left( \frac{1+\beta}{1+3\beta} \sqrt{3(1+\gamma)^{3/\gamma}} 2^{n-1+1/s} \right)^s - 1 \right]^{1/s}.$$

(c) The proof of Part (c) follows from [49].

(d) The proof of lower bound follows from Parts (b) and (c).

For the upper bound, by Part (a), we have

$$1 = \int_{\mathbb{R}^n} f(x) dx \geq \int_{\|x\| \leq d} f(x) dx \geq d^n \text{vol}(B_{n-1})(2 - 2^{-(n+1)s})^{-1/s} f(0),$$

where $\text{vol}(B_{n-1})$ represents the volume of $n-1$-dimensional unit ball. So

$$f(0) \leq \frac{(2 - 2^{-(n+1)s})^{1/s}}{d^n \text{vol}(B_{n-1})} = (2 - 2^{-(n+1)s})^{1/s} \frac{n\Gamma(n/2)}{2\pi^{n/2} d^n}.$$

(e) The proof of (e) follows from Parts (b) and (d).

(f) Define an orthogonal coordinate system in which $\ell$ is the $x_n$-axis. Let $h$ be the marginal of function $f$ over first $n-1$ variables, namely,

$$h(x_1, ..., x_{n-1}) = \int f(x_1, ..., x_{n-1}, x_n) dx_n.$$

Then

$$\int_\ell f = h(0) \leq (2 - 2^{-ns})^{1/s} \frac{(n-1)\Gamma((n-1)/2)}{2\pi^{(n-1)/2} d^{n-1}}.$$

$\square$

# I  Proof of Lemma 10

**Lemma 10** (restated) *Let $f : \mathbb{R}^n \to \mathbb{R}_+$ be an isotropic $s$-concave density. Then $f(x) \leq \beta_1(n, s)(1 - s\beta_2(n, s)\|x\|)^{1/s}$ for every $x \in \mathbb{R}^n$, where*

$$\beta_1(n, s) = (2 - 2^{-(n+1)s})^{\frac{1}{s}} \frac{1}{2\pi^{n/2} d^n} (1-s)^{-\frac{1}{s}} n\Gamma(n/2) \left[ \left( \frac{1+\beta}{1+3\beta} \sqrt{3(1+\gamma)^{3/\gamma}} 2^{n-1+1/s} \right)^s - 1 \right]^{1/s},$$

*and*

$$\beta_2(n, s) = \frac{2\pi^{(n-1)/2} d^{n-1}}{(n-1)\Gamma((n-1)/2)} (2 - 2^{-ns})^{-1/s} \frac{[(a + (1-s)\beta_1(n, s)^s)^{1+1/s} - a^{1+1/s}]s}{\beta_1(n, s)^s (1+s)(1-s)},$$

$$d \;=\; (1 \;+\; \gamma)^{-\frac{1}{\gamma}\frac{1+3\beta}{3+3\beta}}, \qquad \beta \;=\; \frac{s}{1+(n-1)s}, \qquad \gamma \;=\; \frac{\beta}{1+\beta}, \qquad a \;=\;$$
$$(4e\pi)^{-\frac{ns}{2}}\left[\left(\frac{1+\beta}{1+3\beta}\sqrt{3(1+\gamma)^{3/\gamma}}2^{n-1+1/s}\right)^{s}-1\right]^{-1}.$$

*Proof.* We first note that when $\|x\| \le 1/\beta_2$, $\beta_1(1-s\beta_2\|x\|)^{1/s} \ge \beta_1(1-s)^{1/s} \ge f(x)$ by Theorem 9 (e). So the conclusion holds.

We now assume that there is a point $v$ such that $\|v\| > 1/\beta_2$ but $f(v) > \beta_1(1-s\beta_2\|v\|)^{1/s}$. Denote by $[0, v]$ the line segment between the origin $0$ and the point $v$, and let $\ell$ be the line through $v$ and $0$. We will prove that

$$\int_\ell f > (2 - 2^{-ns})^{1/s}\frac{(n-1)\Gamma((n-1)/2)}{2\pi^{(n-1)/2}d^{n-1}},$$

which leads to a contradiction with Theorem 9 (f). Let $x$ be the convex combination of points $0$ and $v$, i.e., $x = (1 - \|x\|/\|v\|)0 + (\|x\|/\|v\|)v$, where $0 \le \|x\| \le \|v\|$. Then by the $s$-concavity of $f$ and Theorem 9 (d),

$$
\begin{aligned}
f(x) &\ge \left[\left(1 - \frac{\|x\|}{\|v\|}\right)f(0)^s + \frac{\|x\|}{\|v\|}f(v)^s\right]^{1/s} \\
&> \left[f(0)^s + \frac{\|x\|}{\|v\|}f(v)^s\right]^{1/s} \\
&> \left[f(0)^s + \frac{\|x\|}{\|v\|}\beta_1^s - s\beta_1^s\beta_2\|x\|\right]^{1/s} \\
&> \left[f(0)^s + (1-s)\beta_1^s\beta_2\|x\|\right]^{1/s} \\
&> \left\{(4e\pi)^{-ns/2}\left[\left(\frac{1+\beta}{1+3\beta}\sqrt{3(1+\gamma)^{3/\gamma}}2^{n-1+1/s}\right)^{s}-1\right]^{-1} + (1-s)\beta_1^s\beta_2\|x\|\right\}^{1/s}.
\end{aligned}
$$

Thus

$$
\begin{aligned}
\int_\ell f \ge \int_{[0,v]} f &= \int_0^{\|v\|}\left\{(4e\pi)^{-ns/2}\left[\left(\frac{1+\beta}{1+3\beta}\sqrt{3(1+\gamma)^{3/\gamma}}2^{n-1+1/s}\right)^{s}-1\right]^{-1} + (1-s)\beta_1^s\beta_2 r\right\}^{1/s} dr \\
&> \int_0^{1/\beta_2}\left\{(4e\pi)^{-ns/2}\left[\left(\frac{1+\beta}{1+3\beta}\sqrt{3(1+\gamma)^{3/\gamma}}2^{n-1+1/s}\right)^{s}-1\right]^{-1} + (1-s)\beta_1^s\beta_2 r\right\}^{1/s} dr \\
&\ge (2 - 2^{-ns})^{1/s}\frac{(n-1)\Gamma((n-1)/2)}{2\pi^{(n-1)/2}d^{n-1}}.
\end{aligned}
$$

So the proof is completed. $\qquad\square$

## J  Proof of Theorem 11

**Theorem 11** (restated) *Let $\mathcal{D}$ be an isotropic $s$-concave distribution in $\mathbb{R}^n$. Denote by $f_3(s,n) = 2(1+ns)/(1+(n+2)s)$. Then for any unit vector $w$,*

$$\Pr_{x\sim\mathcal{D}}[|w \cdot x| \le t] \le f_3(s,n)t. \tag{3}$$

*Moreover, if $t \le d = \left(\frac{1+2\gamma}{1+\gamma}\right)^{-\frac{1+\gamma}{\gamma}}\frac{1+3\gamma}{3+3\gamma}$ where $\gamma = \frac{s}{1+(n-1)s}$, then*

$$\Pr_{x\sim\mathcal{D}}[|w \cdot x| \le t] > f_2(s,n)t, \tag{4}$$

*where $f_2(s,n) = 2(2 - 2^{-2\gamma})^{-1/\gamma}(4e\pi)^{-1/2}\left(2\left(\frac{1+\gamma}{1+3\gamma}\sqrt{3}\left(\frac{1+2\gamma}{1+\gamma}\right)^{\frac{3+3\gamma}{2\gamma}}\right)^{\gamma} - 1\right)^{-1/\gamma}.$*

*Proof.* Define an orthogonal coordinate system in which $w$ is an axis. Then the distribution of $w \cdot x$ is equal to the first marginal of the distribution $\mathcal{D}$, with isotropic $\gamma = \frac{s}{1+(n-1)s}$-concave density $g$ by Theorem 3. According to the upper bound given by Lemma 7 (a),

$$\Pr_{x \sim \mathcal{D}}[|w \cdot x| \leq t] = \int_{-t}^{t} g(x)dx \leq \frac{1+\gamma}{1+3\gamma} \int_{-t}^{t} dx = 2\frac{1+ns}{1+(n+2)s}t.$$

We now prove the later part of the theorem by a similar argument. By Theorem 9 (a) (d), for 1-dimensional $\gamma$-concave density $f(u)$ and $\|u\| \leq d$, we have

$$f(u) \geq (2 - 2^{-2\gamma})^{-1/\gamma} f(0)$$

$$> (2 - 2^{-2\gamma})^{-1/\gamma} (4e\pi)^{-1/2} \left( 2 \left( \frac{1+\gamma}{1+3\gamma} \sqrt{3} \left( \frac{1+2\gamma}{1+\gamma} \right)^{\frac{3+3\gamma}{2\gamma}} \right)^{\gamma} - 1 \right)^{-1/\gamma}$$

$$\triangleq \frac{f_2(s,n)}{2}.$$

Therefore,

$$\Pr_{x \sim \mathcal{D}}[|w \cdot x| \leq t] = \int_{-t}^{t} g(x)dx > \frac{f_2(s,n)}{2} \int_{-t}^{t} dx = f_2(s,n)t.$$

$\square$

## K   Proof of Theorem 12

**Theorem 12** (restated) *Assume $\mathcal{D}$ is an isotropic $s$-concave distribution in $\mathbb{R}^n$. Then for any two unit vectors $u$ and $v$ in $\mathbb{R}^n$, we have $d_{\mathcal{D}}(u,v) = \Pr_{x \sim \mathcal{D}}[\text{sign}(u \cdot x) \neq \text{sign}(v \cdot x)] \geq f_1(s,n)\theta(u,v)$, where $f_1(s,n) = c(2 - 2^{-3\alpha})^{-\frac{1}{\alpha}} \left[ \left( \frac{1+\beta}{1+3\beta} \sqrt{3(1+\gamma)^{3/\gamma}} 2^{1+1/\alpha} \right)^{\alpha} - 1 \right]^{-\frac{1}{\alpha}} (1+\gamma)^{-2/\gamma} \left( \frac{1+3\beta}{3+3\beta} \right)^2$, $c$ is an absolute constant, $\alpha = \frac{s}{1+(n-2)s}$, $\beta = \frac{s}{1+(n-1)s}$, $\gamma = \frac{s}{1+ns}$.*

*Proof.* Consider the 2-dimensional space spanned by vectors $u$ and $v$, and let $\mathcal{D}_2$ be the marginal of distribution $\mathcal{D}$ over that space. Since $d_{\mathcal{D}}(u,v) = d_{\mathcal{D}_2}(u',v')$, where $u'$ and $v'$ are projection of $u$ and $v$, respectively, we only need to focus on the marginal distribution $\mathcal{D}_2$, which has an $\alpha$-concave density according to Theorem 3, and is isotropic according to Theorem 3.

Let $A$ be the disagreement region of $u$ and $v$ intersected with the ball of radius $d = (1+\gamma)^{-1/\gamma}\frac{1+3\beta}{3+3\beta}$ in $\mathbb{R}^2$. By Theorem 9 (a) and (d), $d_{\mathcal{D}}(u,v) = d_{\mathcal{D}_2}(u',v') \geq \text{vol}(A)\inf_{x \in A} p(x) \geq f_1(s,n)\theta(u',v') = f_1(s,n)\theta(u,v)$, where $p(x)$ is the density of distribution $\mathcal{D}_2$. $\square$

## L   Proof of Theorem 13

**Theorem 13** (restated) *Let $u$ and $v$ be two vectors in $\mathbb{R}^n$ and assume that $\theta(u,v) = \theta < \pi/2$. Let $\mathcal{D}$ be an isotropic $s$-concave distribution. Then for any absolute constant $c_1 > 0$ and any function $f_1(s,n) > 0$, there exists a function $f_4(s,n) > 0$ such that*

$$\Pr_{x \sim \mathcal{D}}[\text{sign}(u \cdot x) \neq \text{sign}(v \cdot x) \text{ and } |v \cdot x| \geq f_4(s,n)\theta] \leq c_1 f_1(s,n)\theta,$$

*where $f_4(s,n) = \frac{4\beta_1(2,\alpha)B(-1/\alpha-3,3)}{-c_1 f_1(s,n)\alpha^3\beta_2(2,\alpha)^3}$, $B(\cdot,\cdot)$ is the beta function, $\alpha = s/(1+(n-2)s)$, $\beta_1(2,\alpha)$ and $\beta_2(2,\alpha)$ are given by Lemma 10.*

*Proof.* Let $E$ be the event that we want to bound. Theorem 3 implies that, without loss of generality, we can focus on the case when $n = 2$. Then the projected distribution $\mathcal{D}_2$ has an $\alpha$-concave density, where $\alpha = \frac{s}{1+(n-2)s}$.

We first claim that each member $x$ of $E$ satisfies $\|x\| \geq f_4$. To see this, without loss of generality, we assume that $v \cdot x$ is positive. Then for any $x \in E$, $u \cdot x < 0$, i.e., $\theta(u,x) \geq \pi/2$. Hence

$\theta(x,v) \geq \theta(u,x) - \theta(u,v) \geq \pi/2 - \theta$. Since $|v \cdot x| \geq f_4\theta$ implies $\|x\|\cos\theta(v,x) \geq f_4\theta$, we must have $\|x\|\cos(\pi/2 - \theta) \geq f_4\theta$, namely, $\|x\| \geq f_4\theta/\sin(\theta) \geq f_4$. Let $\mathsf{ball}(r)$ denote the ball of radius $r$ centered at the origin. This implies that

$$\Pr[E] = \sum_{i=1}^{\infty} \Pr[E \cap (\mathsf{ball}((i+1)f_4) - \mathsf{ball}(if_4))].$$

Denote by $f(x_1, x_2)$ the $\alpha$-concave density function of $\mathcal{D}_2$. For any term $i \geq 1$, by Lemma 10, we have

$$\Pr[E \cap (\mathsf{ball}((i+1)f_4) - \mathsf{ball}(if_4))]$$

$$= \int_{(x_1,x_2)\in\mathsf{ball}((i+1)f_4)-\mathsf{ball}(if_4)} 1_E(x_1, x_2) f(x_1, x_2) dx_1 dx_2$$

$$\leq \int_{(x_1,x_2)\in\mathsf{ball}((i+1)f_4)-\mathsf{ball}(if_4)} 1_E(x_1, x_2)\beta_1(2,\alpha)(1 - \alpha\beta_2(2,\alpha)\|x\|)^{1/\alpha} dx_1 dx_2$$

$$\leq \beta_1(2,\alpha)\left(1 - \alpha\beta_2(2,\alpha)if_4\right)^{1/\alpha} \int_{(x_1,x_2)\in\mathsf{ball}((i+1)f_4)-\mathsf{ball}(if_4)} 1_E(x_1, x_2) dx_1 dx_2$$

$$\leq \beta_1(2,\alpha)\left(1 - \alpha\beta_2(2,\alpha)if_4\right)^{1/\alpha} \int_{(x_1,x_2)\in\mathsf{ball}((i+1)f_4)} 1_E(x_1, x_2) dx_1 dx_2.$$

Denote by $B_1$ the unit ball in $\mathbb{R}^2$. Notice that

$$\int_{(x_1,x_2)\in\mathsf{ball}((i+1)f_4)} 1_E(x_1, x_2) dx_1 dx_2 \leq \mathrm{vol}(B_1)(i+1)^2 f_4^2 \theta/\pi.$$

Thus

$$\Pr[E] \leq \sum_{i=1}^{\infty} \beta_1(2,\alpha)\left(1 - \alpha\beta_2(2,\alpha)if_4\right)^{1/\alpha} \mathrm{vol}(B_1)(i+1)^2 f_4^2 \theta/\pi$$

$$\leq \frac{4f_4^2}{\pi}\mathrm{vol}(B_1)\beta_1(2,\alpha)\theta \sum_{i=1}^{\infty} (1 - \alpha\beta_2(2,\alpha)if_4)^{1/\alpha} i^2$$

$$\leq \frac{4f_4^2}{\pi}\mathrm{vol}(B_1)\beta_1(2,\alpha)\theta \int_0^{\infty} x^2(1 - \alpha\beta_2(2,\alpha)f_4 x)^{1/\alpha} dx$$

$$= \frac{4f_4^2}{\pi}\mathrm{vol}(B_1)\beta_1(2,\alpha)\frac{B(-1/\alpha - 3, 3)}{(-\alpha\beta_2(2,\alpha)f_4)^3} \times \theta$$

$$= 4\beta_1(2,\alpha)\frac{B(-1/\alpha - 3, 3)}{-\alpha^3\beta_2(2,\alpha)^3 f_4} \times \theta.$$

Choosing $f_4(s,n) = \frac{4\beta_1(2,\alpha)B(-1/\alpha-3,3)}{-c_1 f_1(s,n)\alpha^3\beta_2(2,\alpha)^3}$, the proof is completed. $\qquad\square$

## M  Proof of Theorem 14

Before proceeding, we first prove the following lemma which is critical to the proof of Theorem 14.

**Lemma 23.** *For $d$ given by Theorem 9 (a), there exist such that for any isotropic $s$-concave distribution $\mathcal{D}$, for any $a$ such that $\|a\| \leq 1$ and $\|u - a\| \leq r$, for any $0 < t \leq d$, and for any $K \geq 4$, we have*

$$\Pr_{X\sim\mathcal{D}_{u,t}}(|a \cdot x| > K\sqrt{r^2 + t^2}) \leq \frac{4\beta_1(2,\eta)}{f_2(s,n)\beta_2(2,\eta)}\frac{1}{\eta+1}\left(1 - c\eta\beta_2(2,\eta)K\sqrt{1 + \left(\frac{t}{r}\right)^2}\right)^{\frac{\eta+1}{\eta}},$$

*where $\beta_1(2,\eta)$, $\beta_2(2,\eta)$, and $Q(\gamma)$, are given by Lemma 10 and Theorem 11, respectively, $\eta = \frac{s}{1+(n-2)s}$, and $c$ is an absolute constant.*

*Proof.* Without loss of generality, we assume that $u = (1, 0, ..., 0)$. Let $a' = (0, a_2, ..., a_d)$ and $X = (x_1, x_2, ..., x_d) \sim \mathcal{D}_{u,t}$. So the probability that we want to bound is

$$\Pr_{X \sim \mathcal{D}_{u,t}}(|a \cdot x| > K\sqrt{r^2 + t^2}) = \frac{\Pr_{x \sim \mathcal{D}}(|a \cdot x| > K\sqrt{r^2 + t^2}, |x_1| \leq t)}{\Pr_{x \sim \mathcal{D}}(|x_1| \leq t)}.$$

According to Theorem 11, there is a function $Q(\gamma)$ such that the denominator obeys the following lower bound when $t \leq d$:

$$\Pr_{X \sim \mathcal{D}}(|x_1| \leq t) \geq f_2(s, n)t.$$

So the remainder of the proof is to bound the numerator. Note that we have

$$\Pr_{x \sim \mathcal{D}}(|a \cdot x| > K\sqrt{r^2 + t^2}, |x_1| \leq t) \leq \Pr_{x \sim \mathcal{D}}(|a' \cdot x| > K\sqrt{r^2 + t^2} - t, |x_1| \leq t)$$

$$\leq \Pr_{x \sim \mathcal{D}}(|a' \cdot x| > (K-1)\sqrt{r^2 + t^2}, |x_1| \leq t).$$

Denote by $a'' = \frac{a'}{\|a'\|}$. Define random variable $Y$ as $a'' \cdot x$ and $Z$ as $x_1$ where $x \sim \mathcal{D}$. Then the joint distribution of $Y$ and $Z$ is isotropic $\beta$-concave with $\eta = \frac{s}{1+(n-2)s}$. Let $f(y, z)$ be the density of such a distribution. Then we can bound the numerator by

$$4 \Pr_{x \sim \mathcal{D}}(a' \cdot x > (K-1)\sqrt{r^2 + t^2}, 0 \leq x_1 \leq t) = 4 \Pr_{X \sim \mathcal{D}}\left(a'' \cdot x > \frac{(K-1)\sqrt{r^2 + t^2}}{\|a'\|}, 0 \leq x_1 \leq t\right)$$

$$\leq 4 \int_0^t \int_{\frac{(K-1)\sqrt{r^2+t^2}}{\|a'\|}}^{\infty} f(y, z)dydz.$$

By Lemma 10, we note that

$$f(y, z) \leq \beta_1(2, \eta)(1 - \eta\beta_2(2, \eta)\sqrt{y^2 + z^2})^{1/\eta}.$$

Therefore, the numerator can be upper bounded by

$$4\beta_1(2, \eta) \int_0^t \int_{\frac{(K-1)\sqrt{r^2+t^2}}{\|a'\|}}^{\infty} (1 - \eta\beta_2(2, \eta)\sqrt{y^2 + z^2})^{1/\eta}dydz$$

$$\leq 4\beta_1(2, \eta) \int_0^t \int_{\frac{(K-1)\sqrt{r^2+t^2}}{\|a'\|}}^{\infty} (1 - \eta\beta_2(2, \eta)y)^{1/\eta}dydz \tag{5}$$

$$= 4t\beta_1(2, \eta) \int_{\frac{(K-1)\sqrt{r^2+t^2}}{\|a'\|}}^{\infty} (1 - \eta\beta_2(2, \eta)y)^{1/\eta}dy$$

$$= \frac{4t\beta_1(2, \eta)}{\beta_2(2, \eta)} \frac{1}{\eta + 1} \left(1 - \eta\beta_2(2, \eta)\frac{(K-1)\sqrt{r^2 + t^2}}{\|a'\|}\right)^{\frac{\eta+1}{\eta}}.$$

Note that $\|a'\| \leq r$. Finally, we have

$$\Pr_{X \sim \mathcal{D}_{u,t}}(|a \cdot x| > K\sqrt{r^2 + t^2}) \leq \frac{4\beta_1(2, \eta)}{f_2(s, n)\beta_2(2, \eta)} \frac{1}{\eta + 1} \left(1 - \eta\beta_2(2, \eta)\frac{(K-1)\sqrt{r^2 + t^2}}{r}\right)^{\frac{\eta+1}{\eta}}$$

$$\leq \frac{4\beta_1(2, \eta)}{f_2(s, n)\beta_2(2, \eta)} \frac{1}{\eta + 1} \left(1 - c\eta\beta_2(2, \eta)\frac{K\sqrt{r^2 + t^2}}{r}\right)^{\frac{\eta+1}{\eta}},$$

for an absolute constant $c$. $\qquad\square$

**Theorem 14** (restated) *Assume that $\mathcal{D}$ is isotropic $s$-concave. For $d$ given by Theorem 9 (a), there is an absolute $C_0$ such that for all $0 < t \leq d$ and for all $a$ such that $\|u - a\| \leq r$ and $\|a\| \leq 1$, $\mathbb{E}_{X \sim \mathcal{D}_{u,t}}[(a \cdot x)^2] \leq f_5(s, n)(r^2 + t^2)$, where $f_5(s, n) = 16 + C_0 \frac{8\beta_1(2,\eta)B(-1/\eta-3,2)}{f_2(s,n)\beta_2(2,\eta)^3(\eta+1)\eta^2}$, $(\beta_1(2, \eta), \beta_2(2, \eta))$ and $f_2(s, n)$ are given by Lemma 10 and Theorem 11, respectively, and $\eta = \frac{s}{1+(n-2)s}$.*

*Proof.* Denote by $z = \sqrt{r^2 + t^2}$. Then we have

$$
\begin{aligned}
\mathbb{E}_{x \sim \mathcal{D}_{u,t}}[(a \cdot x)^2] &= \int_0^\infty \Pr_{x \sim \mathcal{D}_{u,t}}[(a \cdot x)^2 \geq z] dz \\
&\leq 16z^2 + \int_{16z^2}^\infty \Pr_{x \sim \mathcal{D}_{u,t}}[(a \cdot x)^2 \geq z] dz \\
&\leq 16z^2 + \frac{4\beta_1(2,\eta)}{f_2(s,n)\beta_2(2,\eta)} \frac{1}{\eta+1} \int_0^\infty \left(1 - \eta\beta_2(2,\eta)\frac{c\sqrt{z}}{r}\right)^{\frac{\eta+1}{\eta}} dz \\
&= 16z^2 + \frac{8\beta_1(2,\eta)}{f_2(s,n)\beta_2(2,\eta)} \frac{1}{\eta+1} \int_0^\infty y \left(1 - \eta\beta_2(2,\eta)\frac{cy}{r}\right)^{\frac{\eta+1}{\eta}} dy \quad (6) \\
&= 16z^2 + \frac{8\beta_1(2,\eta)}{f_2(s,n)\beta_2(2,\eta)} \frac{1}{\eta+1} C_0 B(-1/\eta - 3, 2) \frac{r^2}{\eta^2\beta_2(2,\eta)^2} \\
&= \left(16 + C_0 \frac{8\beta_1(2,\eta)B(-1/\eta - 3, 2)}{f_2(s,n)\beta_2(2,\eta)^3(\eta+1)\eta^2}\right) r^2 + 16t^2 \\
&\leq \left(16 + C_0 \frac{8\beta_1(2,\eta)B(-1/\eta - 3, 2)}{f_2(s,n)\beta_2(2,\eta)^3(\eta+1)\eta^2}\right) (r^2 + t^2),
\end{aligned}
$$

where $c, C_0$ are absolute constants. $\qquad\square$

## N  Proof of Theorem 15

**Theorem 15** (restated) *In the realizable case, let $\mathcal{D}$ be an isotropic s-concave distribution in $\mathbb{R}^n$. There exist constants $C$ and $c$ such that for any $0 < \epsilon < 1/4$ and $\delta > 0$, Algorithm 2 with $b_k = \min\{\Theta(2^{-k}f_4 f_1^{-1}), d\}$ and $m_k = C\left(\frac{f_3 b_{k-1}}{2^{-k}}\left(n\log\frac{f_3 b_{k-1}}{2^{-k}} + \log\frac{1+s-k}{\delta}\right)\right)$, after $T = \lceil \log\frac{1}{c\epsilon}\rceil$ iterations, outputs a linear separator of error at most $\epsilon$ with probability at least $1 - \delta$.*

*Proof.* We will show by induction that for all $k \leq s$, with probability at least $1 - \frac{\delta}{2}\sum_{i<k}\frac{1}{(1+s-i)^2}$, any $w$ that is consistent with the examples in $W(k)$, e.g. $w_k$, has error at most $c2^{-k}$.

The case of $k = 1$ follows from the VC theory (Theorem 30). Assume now that the claim is true for $k - 1$. We now consider the $k$th iteration. Denote by $S_{k-1} = \{x : |w_{k-1} \cdot x| \leq b_{k-1}\}$ and $\bar{S}_{k-1} = \{x : |w_{k-1} \cdot x| > b_{k-1}\}$. By the induction hypothesis, with probability at least $1 - \frac{\delta}{2}\sum_{i<k-1}\frac{1}{(1+s-i)^2}$, any $w$ that is consistent with $W(k-1)$, including $w_{k-1}$, has error at most $c2^{-(k-1)}$. For such a hypothesis $w$ and $w_{k-1}$, by Theorem 12, we have $\theta(w, w^*) \leq cf_1^{-1}2^{-(k-1)}$ and $\theta(w_{k-1}, w^*) \leq cf_1^{-1}2^{-(k-1)}$. Thus $\theta(w_{k-1}, w) \leq \theta(w_{k-1}, w^*) + \theta(w^*, w) \leq 4cf_1^{-1}2^{-k}$. So by Theorem 13, there is a choice of band width $b_{k-1} = \min\{\Theta(f_4 f_1^{-1}2^{-k}), d\}$ such that $\Pr(\text{sign}(w \cdot x) \neq \text{sign}(w_{k-1} \cdot x), x \in \bar{S}_{k-1}) \leq \frac{c2^{-k}}{4}$ and $\Pr[\text{sign}(w_{k-1} \cdot x) \neq \text{sign}(w^* \cdot x), x \in \bar{S}_{k-1}] \leq \frac{c2^{-k}}{4}$. Therefore, $\Pr[\text{sign}(w \cdot x) \neq \text{sign}(w^* \cdot x), x \in \bar{S}_{k-1}] \leq \frac{c2^{-k}}{2}$.

We now consider the case when $x \in S_{k-1}$. By Algorithm 2, we label $m_k$ data points in $S_{k-1}$ at the $(k-1)$th iteration. So according to the VC theory (Theorem 30), with probability at least $1 - \delta/(4(1+s-k)^2)$, for all $w$ that is consistent with the examples in $W(k)$, $\text{err}(w|S_{k-1}) = \Pr[\text{sign}(w \cdot x) \neq \text{sign}(w^* \cdot x)|x \in S_{k-1}] \leq \frac{c2^{-k}}{2b_{k-1}f_3}$. Finally, note that Theorem 11 implies that $\Pr(S_{k-1}) \leq f_3 b_{k-1}$. So we have $\text{err}(w) = \Pr[\text{sign}(w \cdot x) \neq \text{sign}(w^* \cdot x), x \in \bar{S}_{k-1}] + \Pr[\text{sign}(w \cdot x) \neq \text{sign}(w^* \cdot x), x \in S_{k-1}] \leq \frac{c2^{-k}}{2} + \frac{c2^{-k}}{2b_{k-1}f_3} \times f_3 b_{k-1} = c2^{-k}$. The proof is completed. $\qquad\square$

## O  Proof of Theorem 16

Before proceeding, let $\ell_\tau(w, x, y) = \max\{0, 1 - y(w \cdot x)/\tau\}$, $\ell_\tau(w, T) = \frac{1}{|T|}\sum_{(x,y)\in T}\ell_\tau(w, x, y)$, and $L_\tau(w, \mathcal{D}) = \mathbb{E}_{x\sim\mathcal{D}}\ell_\tau(w, x, \text{sign}(w^* \cdot x))$. Our analysis will involve the distribution $\mathcal{D}_{w,t}$ obtained by conditioning $\mathcal{D}$ on membership in the band, namely, the set $\{x : |w \cdot x| \leq t\}$.

**Lemma 24.** $L_{\tau_k}(w^*, \mathcal{D}_{w_{k-1}, b_{k-1}}) \le \kappa/6$, if $\kappa \ge \frac{6 f_3 \tau_k}{f_2 b_{k-1}}$ and $b_{k-1} \le d$.

*Proof.* Note that $y(w^* \cdot x)$ cannot be negative on any clean example $(x, y)$. So we have $\ell(w^*, x, y) = \max\{0, 1 - y(w^* \cdot x)/\tau_k\} \le 1$ and $w^*$ pays a non-zero hinge loss only inside the margin $\{x : |w^* \cdot x| \le \tau_k\}$. Thus $L_{\tau_k}(w^*, \mathcal{D}_{w_{k-1}, b_{k-1}}) \le \Pr_{\mathcal{D}_{w_{k-1}, b_{k-1}}}(|w^* \cdot x| \le \tau_k) = \Pr_{\mathcal{D}}(|w^* \cdot x| \le \tau_k, |w_{k-1} \cdot x| \le b_{k-1}) / \Pr_{\mathcal{D}}(|w_{k-1} \cdot x| \le b_{k-1})$. Notice that the numerator can be bounded by $\Pr_{\mathcal{D}}(|w^* \cdot x| \le \tau_k) \le f_3 \tau_k$ according to Theorem 11. As for the denominator, by Theorem 11 we have $\Pr_{\mathcal{D}}(|w_{k-1} \cdot x| \le b_{k-1}) \ge f_2 \min\{b_{k-1}, d\}$. So we have $L_{\tau_k}(w^*, \mathcal{D}_{w_{k-1}, b_{k-1}}) \le f_3 \tau_k / (f_2 \min\{b_{k-1}, d\}) \le \kappa/6$. $\square$

Let $\widetilde{\mathcal{P}}_k$ be the noisy distribution of $(x, y)$ where $x \sim \mathcal{D}_{w_{k-1}, b_{k-1}}$ and $y$ obeys the adversarial noise model, and denote by $\mathcal{P}_k$ the clean distribution of $(x, y)$ where $x \sim \mathcal{D}_{w_{k-1}, b_{k-1}}$ and $y = \mathrm{sign}(w^* \cdot x)$. The following key lemma bounds the distance of expected loss w.r.t. the distributions $\widetilde{\mathcal{P}}_k$ and $\mathcal{P}_k$.

**Lemma 25.** *There exists an absolute constant $c$ such that for any $w \in \mathsf{ball}(w_{k-1}, r_k)$, we have that*

$$\left| \mathbb{E}_{(x,y) \sim \mathcal{P}_k} \ell(w, x, y) - \mathbb{E}_{(x,y) \sim \widetilde{\mathcal{P}}_k} \ell(w, x, y) \right| \le \frac{2}{\tau_k} \sqrt{\frac{\eta f_5 (r_k^2 + b_{k-1}^2)}{f_2 b_{k-1}}}.$$

*Proof.* Denote by $N$ the set of noisy examples. Let $\widetilde{\mathcal{P}}$ be the noisy distribution of $(x, y)$ where $x \sim \mathcal{D}$ and $y$ obeys the adversarial noise model. We have

$$\left| \mathbb{E}_{(x,y) \sim \widetilde{\mathcal{P}}_k} [\ell_{\tau_k}(w^*, x, y)] - \mathbb{E}_{(x,y) \sim \mathcal{P}_k} [\ell_{\tau_k}(w^*, x, y)] \right|$$

$$\le \left| \mathbb{E}_{(x,y) \sim \widetilde{\mathcal{P}}_k} [\mathbf{1}_{x \in N} (\ell_{\tau_k}(w^*, x, y) - \ell_{\tau_k}(w^*, x, \mathrm{sign}(w^* \cdot x)))] \right|$$

$$\le 2 \, \mathbb{E}_{(x,y) \sim \widetilde{\mathcal{P}}_k} \left[ \mathbf{1}_{x \in N} \left( \frac{|w^* \cdot x|}{\tau_k} \right) \right]$$

$$\le \frac{2}{\tau_k} \sqrt{\Pr_{(x,y) \sim \widetilde{\mathcal{P}}_k} [x \in N]} \times \sqrt{\mathbb{E}_{(x,y) \sim \widetilde{\mathcal{P}}_k} [(w^* \cdot x)^2]} \qquad \text{(By Cauchy Schwarz)}$$

$$\le \frac{2}{\tau_k} \sqrt{\frac{\eta}{\Pr_{\widetilde{\mathcal{P}}}(|w_{k-1} \cdot x| \le b_{k-1})}} \times \sqrt{f_5(r_k^2 + b_{k-1}^2)} \qquad \text{(By Theorem 14)}$$

$$\le \frac{2}{\tau_k} \sqrt{\frac{\eta f_5(r_k^2 + b_{k-1}^2)}{f_2 b_{k-1}}}. \qquad \text{(By Theorem 11)}$$

$\square$

**Lemma 26.** *Denote by $W$ the samples drawn from the noisy distribution $\widetilde{\mathcal{P}}_k$ and suppose that $|W| = O\left( \frac{[b_{k-1}s + \tau_k(1+ns)\sqrt{n}[1-(\delta/(\sqrt{n}(k+k^2)))^{s/(1+ns)}] + \tau_k s]^2}{\kappa^2 \tau_k^2 s^2} \left( n + \log \frac{k+k^2}{\delta} \right) \right)$. Then with probability at least $1 - \frac{\delta}{k+k^2}$, for all $w \in \mathsf{ball}(w_{k-1}, r_k)$, we have*

$$\left| \mathbb{E}_{(x,y) \sim \widetilde{\mathcal{P}}_k} \ell(w, x, y) - \ell(w, W) \right| \le \kappa/16.$$

*Proof.* To establish the lemma, we apply some standard VC tools (Theorem 31). Note that the pseudo-dimension of $\{\ell(w, \cdot) : w \in \mathbb{R}^n\}$ is at most $n$ [5]. To use Theorem 31, we first provide the upper bound on the loss. On one hand, note that

$$\ell(w, x, y) \le 1 + \frac{|w \cdot x|}{\tau_k} \le 1 + \frac{|w_{k-1} \cdot x| + \|w - w_{k-1}\| \|x\|}{\tau_k}$$

$$\le 1 + \frac{b_{k-1} + \tau_k \|x\|}{\tau_k}.$$

On the other hand, by Theorem 5 and the union bound, with probability at least $1 - \frac{\delta}{k+k^2}$, we have that $\max_{x \in W} \|x\| \le C \frac{(1+ns)\sqrt{n}}{s} \left\{ 1 - \left[ \frac{\delta}{6(k+k^2)|W|} \right]^{s/(1+ns)} \right\}$, for an absolute constant $C$. The conclusion then follows from Theorem 31. $\square$

**Lemma 27.** *Let* $k \leq \lceil \log(1/(c\epsilon)) \rceil$ *where* $c$ *is an absolute constant. If* $\kappa = \max\left\{ \frac{f_3 \tau_k}{f_2 \min\{b_{k-1}, d\}}, \frac{b_{k-1}\sqrt{f_5}}{\tau_k \sqrt{f_2}} \right\}$, $r_k \leq O(b_{k-1})$, $\eta \leq O(b_{k-1})$, $m_k = O\left( \frac{[b_{k-1}s + \tau_k(1+ns)\sqrt{n}[1-(\delta/(k+k^2))^{s/(1+ns)}] + \tau_k s]^2}{\kappa^2 \tau_k^2 s^2} \left( n + \log \frac{k+k^2}{\delta} \right) \right)$, *and* $b_{k-1} \leq d$, *then with probability at least* $1 - \frac{\delta}{k+k^2}$, *we have* $\mathsf{err}_{\mathcal{D}_{w_{k-1}, b_{k-1}}}(w_k) \leq \kappa$.

*Proof.* With probability at least $1 - \frac{\delta}{k+k^2}$, we have

$$
\begin{aligned}
\mathsf{err}_{\mathcal{D}_{w_{k-1}, b_{k-1}}}(w_k) &= \mathsf{err}_{\mathcal{D}_{w_{k-1}, b_{k-1}}}(v_k) \\
&\leq \mathbb{E}_{(x,y)\sim\mathcal{P}_k} \ell(v_k, x, y) \\
&\leq \mathbb{E}_{(x,y)\sim\widetilde{\mathcal{P}}_k} \ell(v_k, x, y) + \frac{2}{\tau_k} \sqrt{\frac{\eta f_5(r_k^2 + b_{k-1}^2)}{f_2 b_{k-1}}} && \text{(By Lemma 25)} \\
&\leq \ell(v_k, W) + \frac{2}{\tau_k} \sqrt{\frac{\eta f_5(r_k^2 + b_{k-1}^2)}{f_2 b_{k-1}}} + \frac{\kappa}{16} && \text{(By Lemma 26)} \\
&\leq \ell(w^*, W) + \frac{4}{\tau_k} \sqrt{\frac{\eta f_5(r_k^2 + b_{k-1}^2)}{f_2 b_{k-1}}} + \frac{\kappa}{8} && \text{(Since } \|v_k\| \geq 1/2 \text{)} \\
&\leq \mathbb{E}_{(x,y)\sim\widetilde{\mathcal{P}}_k} \ell(w^*, x, y) + \frac{4}{\tau_k} \sqrt{\frac{\eta f_5(r_k^2 + b_{k-1}^2)}{f_2 b_{k-1}}} + \frac{\kappa}{4} && \text{(By Lemma 26)} \\
&\leq \mathbb{E}_{(x,y)\sim\mathcal{P}_k} \ell(w^*, x, y) + \frac{6}{\tau_k} \sqrt{\frac{\eta f_5(r_k^2 + b_{k-1}^2)}{f_2 b_{k-1}}} + \frac{\kappa}{4} && \text{(By Lemma 25)} \\
&\leq \frac{6}{\tau_k} \sqrt{\frac{\eta f_5(r_k^2 + b_{k-1}^2)}{f_2 b_{k-1}}} + \frac{\kappa}{2} && \text{(By Lemma 24)} \\
&\leq \kappa,
\end{aligned}
$$

where the last inequality holds because $\kappa \tau_k \sqrt{\frac{f_2}{f_5}} \geq \Theta(b_{k-1})$, $r_k \leq O(b_{k-1})$, and $\eta \leq O(b_{k-1})$. $\quad\square$

Now we are ready to prove Theorem 16.

**Theorem 16** (restated) *Let* $\mathcal{D}$ *be an isotropic* $s$-concave distribution in $\mathbb{R}^n$ *and the label* $y$ *obeys the adversarial noise model. If the rate* $\eta$ *of adversarial noise satisfies* $\eta < c_0 \epsilon$ *for some absolute constant* $c_0$, *then there exists an absolute constant* $c$ *such that for any* $0 < \epsilon < 1/4$ *and* $\delta > 0$, *Algorithm 1 with* $b_k = \min\{\Theta(2^{-k} f_4 f_1^{-1}), d\}$, $\tau_k = \Theta\left( f_1^{-2} f_2^{-1/2} f_3 f_4^2 f_5^{1/2} 2^{-(k-1)} \right)$, $r_k = \Theta(2^{-k} f_1^{-1})$, $m_k = O\left( \frac{[b_{k-1}s + \tau_k(1+ns)\sqrt{n}[1-(\delta/(k+k^2))^{s/(1+ns)}] + \tau_k s]^2}{\kappa^2 \tau_k^2 s^2} \left( n + \log \frac{k+k^2}{\delta} \right) \right)$, *and* $\kappa = \max\left\{ \frac{f_3 \tau_k}{f_2 \min\{b_{k-1}, d\}}, \frac{b_{k-1}\sqrt{f_5}}{\tau_k \sqrt{f_2}} \right\}$, *after* $T = \lceil \log \frac{1}{c\epsilon} \rceil$ *iterations, outputs a linear separator* $w_T$ *such that* $\mathrm{Pr}_{x\sim\mathcal{D}}[\mathrm{sign}(w_T \cdot x) \neq \mathrm{sign}(w^* \cdot x)] \leq \epsilon$ *with probability at least* $1 - \delta$.

*Proof.* The case of $k = 1$ is obvious. Assume now that the claim is true for $k - 1$. We now consider the $k$th iteration. Denote by $S_{k-1} = \{x : |w_{k-1} \cdot x| \leq b_{k-1}\}$ and $\bar{S}_{k-1} = \{x : |w_{k-1} \cdot x| > b_{k-1}\}$. By the induction hypothesis, with probability at least $1 - \frac{\delta}{2} \sum_{i<k-1} \frac{1}{(1+s-i)^2}$, $w_{k-1}$ has error at most $c2^{-(k-1)}$. Then by Theorem 12, we have $\theta(w_{k-1}, w^*) \leq cf_1^{-1} 2^{-(k-1)}$. On the other hand, since $\|w_{k-1}\| = 1$ and $v_k \in B(w_{k-1}, r_k)$, we have $\theta(w_{k-1}, v_k) \leq r_k$. This in turn implies $\theta(w_{k-1}, w_k) \leq 2^{-k} f_1^{-1}$. So by Theorem 13, there is a choice of band width $2b_{k-1} = O(f_4 f_1^{-1} 2^{-k})$ such that $\mathrm{Pr}(\mathrm{sign}(w_k \cdot x) \neq \mathrm{sign}(w_{k-1} \cdot x), x \in \bar{S}_{k-1}) \leq \frac{c2^{-k}}{4}$ and $\mathrm{Pr}[\mathrm{sign}(w_{k-1} \cdot x) \neq \mathrm{sign}(w^* \cdot x), x \in \bar{S}_{k-1}] \leq \frac{c2^{-k}}{4}$. Therefore, $\mathrm{Pr}[\mathrm{sign}(w_k \cdot x) \neq \mathrm{sign}(w^* \cdot x), x \in \bar{S}_{k-1}] \leq \frac{c2^{-k}}{2}$. Finally, note that Theorem 11 implies that $\mathrm{Pr}(S_{k-1}) \leq f_3 b_{k-1}$. So we have $\mathsf{err}_{\mathcal{D}}(w_k) = \mathrm{Pr}[\mathrm{sign}(w_k \cdot x) \neq$

$\text{sign}(w^* \cdot x), x \in \bar{S}_{k-1}] + \Pr[\text{sign}(w_k \cdot x) \neq \text{sign}(w^* \cdot x), x \in S_{k-1}] = \Pr[\text{sign}(w_k \cdot x) \neq \text{sign}(w^* \cdot x), x \in \bar{S}_{k-1}] + \text{err}_{\mathcal{D}_{w_{k-1}, b_{k-1}}}(w_k) \Pr(x \in S_{k-1}) \leq \frac{c2^{-k}}{2} + \kappa \times f_3 b_{k-1} \leq c2^{-k} = \epsilon$, where the penultimate inequality follows from Lemma 27. The proof is completed. $\square$

# P  Proof of Theorem 17

**Theorem 17** (restated) *Let $\mathcal{D}$ be an isotropic s-concave distribution over $\mathbb{R}^n$. Then for any $w^* \in \mathbb{R}^n$ and $r > 0$, the disagreement coefficient is $\Theta_{w^*, \mathcal{D}}(\epsilon) = O\left(\sqrt{n} \frac{(1+ns)^2}{s(1+(n+2)s)f_1(s,n)}(1 - \epsilon^{s/(1+ns)})\right)$, where $f_1(s,n)$ is given by Theorem 12. In particular, when $s \to 0$ (a.k.a. log-concave), $\Theta_{w^*, \mathcal{D}}(\epsilon) = O(\sqrt{n}\log(1/\epsilon))$.*

*Proof.* Consider any unit $w$ such that $d_{\mathcal{D}}(w, w^*) \leq r$. According to Theorem 12, we have $\|w - w^*\| < \theta(w, w^*) \leq d_{\mathcal{D}}(w, w^*)/f(s) \leq r/f(s)$. Thus for any $x$ such that $\|x\| \leq O(\sqrt{n}\frac{1+ns}{s}(1 - r^{s/(1+ns)}))$, we have $w \cdot x - w^* \cdot x \leq \|w - w^*\| \|x\| < r\sqrt{n}\frac{1+ns}{sf(s)}(1 - r^{s/(1+ns)})$. So as soon as $|w^* \cdot x| \geq r\sqrt{n}\frac{1+ns}{sf(s)}(1 - r^{s/(1+ns)})$, we will have $\text{sign}(w \cdot x) = \text{sign}(w^* \cdot x)$, i.e., $w$ and $w^*$ agree with each other. We now evaluate the probability. By Theorem 11, $\Pr_{x \sim \mathcal{D}}\left[|w^* \cdot x| \leq r\sqrt{n}\frac{1+ns}{sf(s)}(1 - r^{s/(1+ns)})\right] \leq 2\frac{1+ns}{1+(n+2)s}r\sqrt{n}\frac{1+ns}{sf(s)}(1 - r^{s/(1+ns)})$. Moreover, $\Pr_{x \sim \mathcal{D}}\left[\|x\| \geq c\sqrt{n}\frac{1+ns}{s}(1 - r^{s/(1+ns)})\right] \leq Cr$ by Theorem 5. Thus

$$\text{cap}_{w^*, \mathcal{D}}(r) \leq \frac{\Pr_{x \sim \mathcal{D}}\left[|w^* \cdot x| \leq r\sqrt{n}\frac{1+ns}{sf(s)}(1 - r^{s/(1+ns)})\right]}{r} + \frac{\Pr_{x \sim \mathcal{D}}\left[\|x\| \geq c\sqrt{n}\frac{1+ns}{s}(1 - r^{s/(1+ns)})\right]}{r}$$

$$= O\left(\sqrt{n}\frac{(1+ns)^2}{s(1+(n+2)s)f(s)}(1 - r^{s/(1+ns)})\right).$$

Therefore, $\Theta_{w^*, \mathcal{D}}(\epsilon) = \sup_{r \geq \epsilon}[\text{cap}_{w^*, \mathcal{D}}(r)] = O\left(\sqrt{n}\frac{(1+ns)^2}{s(1+(n+2)s)f(s)}(1 - \epsilon^{s/(1+ns)})\right)$. $\square$

# Q  Proof of Theorem 18

**Lemma 28.** *Denote by $R$ the intersections of three origin-centered halfspaces in $\mathbb{R}^n$. Suppose that the instance $x$ in $\mathbb{R}^n$ is drawn from an isotropic s-concave distribution. Then $\Pr[x \in -R] \leq K \Pr[x \in R]$, where $K = \beta_1(3, \kappa)\frac{B(-1/\kappa - 3, 3)}{(-\kappa\beta_2(3,\kappa))^3}\frac{3 + 1/\kappa}{h(\kappa)d^{3+1/\kappa}}$, $\beta_1(3, \kappa)$, $\beta_2(3, \kappa)$, and $a(3, \kappa)$ are as in Lemma 10, $h(\kappa) = \left(\frac{1}{d}((2 - 2^{-4\kappa})^{-1} - 1) + 1\right)^{1/\kappa} (4e\pi)^{-3/2}\left[\left(\frac{1+\beta}{1+3\beta}\sqrt{3(1+\gamma)^{3/\gamma}}2^{2+1/\kappa}\right)^\kappa - 1\right]^{-1/\kappa}$, $d = (1+\gamma)^{-1/\gamma}\frac{1+3\beta}{3+3\beta}$, $\beta = \frac{\kappa}{1+2\kappa}$, $\gamma = \frac{\kappa}{1+\kappa}$, and $\kappa = s/(1 + (n-3)s)$.*

*Proof.* Let $u_1$, $u_2$, and $u_3$ be normals to the hyperplanes bounding the region $R$, namely $R = \{x \in \mathbb{R}^n : u_1 \cdot x \geq 0 \text{ and } u_2 \cdot x \geq 0 \text{ and } u_3 \cdot x \geq 0\}$. Denote by $U$ the linear span of vectors $u_1$, $u_2$, and $u_3$, and let $(e_1, e_2, e_3)$ be an orthogonal basis of $U$ and $(e_1, e_2, e_3, ..., e_n)$ be an extension of basis $(e_1, e_2, e_3)$ to $\mathbb{R}^n$. Represent the components of $x$, $u_1$, $u_2$, and $u_3$ in term of basis $(e_1, e_2, e_3, ..., e_n)$ as

$$x = (x_1, x_2, x_3, x_4, ..., x_n),$$
$$u_1 = (u_{1,1}, u_{1,2}, u_{1,3}, 0, ..., 0),$$
$$u_2 = (u_{2,1}, u_{2,2}, u_{2,3}, 0, ..., 0),$$
$$u_3 = (u_{3,1}, u_{3,2}, u_{3,3}, 0, ..., 0).$$

Denote by $\text{proj}_U(x) \triangleq (x_1, x_2, x_3)$ the projection of $x$ onto subspace $U$, and let $\text{proj}_U(R)$ be the projection of $R$ onto $U$. Because the dot products of a point with normal vectors of $R$ are all that is needed to determine the membership in $R$, we have

$$x \in R \Leftrightarrow u_{j,1}x_1 + u_{j,2}x_2 + u_{j,3}x_3 \geq 0 \text{ for all } j \in \{1, 2, 3\}$$
$$\Leftrightarrow \text{proj}_U(x) \in \text{proj}_U(R). \tag{7}$$

Let $f$ be the density of the isotropic $s$-concave distribution and $g$ be the marginal density of $f$ w.r.t. $(x_1, x_2, x_3)$. Thus by (7),

$$\Pr[x \in R] = \int \cdots \int_R f(x_1, x_2, x_3, x_4, ..., x_n) dx_1 ... dx_n$$

$$= \int \int \int_{\mathsf{proj}_U(R)} g(x_1, x_2, x_3) dx_1 dx_2 dx_3.$$

Note that $f$ is isotropic and $s$-concave. So according to Theorem 3, $g$ is isotropic and $\kappa$-concave with $\kappa = s/(1 + (n-3)s)$. We now use Theorem 9 and Lemma 10 to bound $g$. Specifically, let $u \triangleq (x_1, x_2, x_3)$. On one hand, according to Theorem 9 (a) and (d), for any $u \in \mathbb{R}^3$ such that $\|u\| \leq d$,

$$g(u) \geq \left( \frac{\|u\|}{d} ((2 - 2^{-4\kappa})^{-1} - 1) + 1 \right)^{1/\kappa} f(0)$$

$$> \left( \frac{\|u\|}{d} ((2 - 2^{-4\kappa})^{-1} - 1) + 1 \right)^{1/\kappa} (4e\pi)^{-3/2} \left[ \left( \frac{1+\beta}{1+3\beta} \sqrt{3(1+\gamma)^{3/\gamma}} 2^{2+1/\kappa} \right)^{\kappa} - 1 \right]^{-1/\kappa}$$

$$\triangleq \|u\|^{1/\kappa} h(\kappa),$$

where $d = (1+\gamma)^{-1/\gamma} \frac{1+3\beta}{3+3\beta}$, $\beta = \frac{\kappa}{1+2\kappa}$, $\gamma = \frac{\kappa}{1+\kappa}$, and

$$h(\kappa) = \left( \frac{1}{d} ((2 - 2^{-4\kappa})^{-1} - 1) + 1 \right)^{1/\kappa} (4e\pi)^{-3/2} \left[ \left( \frac{1+\beta}{1+3\beta} \sqrt{3(1+\gamma)^{3/\gamma}} 2^{2+1/\kappa} \right)^{\kappa} - 1 \right]^{-1/\kappa}.$$

On the other hand, by Lemma 10, for every $u \in \mathbb{R}^3$,

$$g(u) \leq \beta_1(3, \kappa)(1 - \kappa\beta_2(3, \kappa)\|u\|)^{1/\kappa},$$

where

$$\beta_1(3, \kappa) = (2 - 2^{-4\kappa})^{1/\kappa} \frac{1}{2\pi^{3/2}d^3} (1 - \kappa)^{-1/\kappa} 3\Gamma(3/2) \left[ \left( \frac{1+\beta}{1+3\beta} \sqrt{3(1+\gamma)^{3/\gamma}} 2^{2+1/\kappa} \right)^{\kappa} - 1 \right]^{1/\kappa},$$

$$\beta_2(3, \kappa) = \frac{2\pi d^2}{2} (2 - 2^{-3s})^{-1/s} \frac{[(a + (1-s)\beta_1(3, \kappa)^{\kappa})^{1+1/\kappa} - a^{1+1/\kappa}]\kappa}{\beta_1(3, \kappa)^s (1 + \kappa)(1 - \kappa)},$$

and

$$a = (4e\pi)^{-3\kappa/2} \left[ \left( \frac{1+\beta}{1+3\beta} \sqrt{3(1+\gamma)^{3/\gamma}} 2^{2+1/\kappa} \right)^{\kappa} - 1 \right]^{-1}.$$

Denote by $R' = \mathsf{proj}_U(R) \cap \mathsf{ball}(0, d)$, and $\mathsf{ball}(0, d)$ is the origin-centered ball of radius $d$ in $\mathbb{R}^3$. Thus we have

$$\int \int \int_{R'} \|u\|^{1/\kappa} h(\kappa) du_1 du_2 du_3 \leq \Pr[x \in R]$$

$$\leq \int \int \int_{\mathsf{proj}_U(R)} \beta_1(3, \kappa)(1 - \kappa\beta_2(3, \kappa)\|u\|)^{1/\kappa} du_1 du_2 du_3.$$

Let $A \triangleq \int \int_{\mathsf{proj}_U(R) \cap \mathbb{S}^2} sin\theta d\varphi d\theta$. Note that

$$\int \int \int_{R'} \|u\|^{1/\kappa} h(\kappa) du_1 du_2 du_3 = A \int_0^d r^2 r^{1/\kappa} h(\kappa) dr = Ah(\kappa) \frac{1}{3 + 1/\kappa} d^{3+1/\kappa},$$

and

$$\int \int \int_{\mathsf{proj}_U(R)} \beta_1(3, \kappa)(1 - \kappa\beta_2(3, \kappa)\|u\|)^{1/\kappa} du_1 du_2 du_3$$

$$= A\beta_1(3, \kappa) \int_0^\infty r^2 (1 - \kappa\beta_2(3, \kappa)r)^{1/\kappa} dr$$

$$= A\beta_1(3, \kappa) \frac{B(-1/\kappa - 3, 3)}{(-\kappa\beta_2(3, \kappa))^3}.$$

So we have

$$Ah(\kappa)\frac{1}{3+1/\kappa}d^{3+1/\kappa} \leq \Pr[x \in R] \leq A\beta_1(3,\kappa)\frac{B(-1/\kappa-3,3)}{(-\kappa\beta_2(3,\kappa))^3},$$

and by symmetry,

$$Ah(\kappa)\frac{1}{3+1/\kappa}d^{3+1/\kappa} \leq \Pr[x \in -R] \leq A\beta_1(3,\kappa)\frac{B(-1/\kappa-3,3)}{(-\kappa\beta_2(3,\kappa))^3}.$$

Therefore,

$$\Pr[x \in -R] \leq \Pr[x \in R]\beta_1(3,\kappa)\frac{B(-1/\kappa-3,3)}{(-\kappa\beta_2(3,\kappa))^3}\frac{3+1/\kappa}{h(\kappa)d^{3+1/\kappa}}.$$

$\square$

**Theorem 18** (restated) *In the PAC realizable case, Algorithm 4 outputs a hypothesis $h$ of error at most $\epsilon$ with probability at least $1 - \delta$ under isotropic s-concave distribution. The label complexity is $M(\epsilon/2, \delta/4, n^2) + \max\{2m_2/\epsilon, (2/\epsilon^2)\log(4/\delta)\}$, where $M(\epsilon, \delta, m)$ is defined by $M(\epsilon, \delta, n) = O\left(\frac{n}{\epsilon}\log\frac{1}{\epsilon} + \frac{1}{\epsilon}\log\frac{1}{\delta}\right)$, $m_2 = M(\max\{\delta/(4eKm_1), \epsilon/2\}, \delta/4, n)$, $K = \beta_1(3,\kappa)\frac{B(-1/\kappa-3,3)}{(-\kappa\beta_2(3,\kappa))^3}\frac{3+1/\kappa}{h(\kappa)d^{3+1/\kappa}}$, $d = (1+\gamma)^{-1/\gamma}\frac{1+3\beta}{3+3\beta}$, $h(\kappa) = \left(\frac{1}{d}((2-2^{-4\kappa})^{-1}-1)+1\right)^{\frac{1}{\kappa}}(4e\pi)^{-\frac{3}{2}}\left[\left(\frac{1+\beta}{1+3\beta}\sqrt{3(1+\gamma)^{3/\gamma}}2^{2+\frac{1}{\kappa}}\right)^{\kappa}-1\right]^{-1/\kappa}$, $\beta = \frac{\kappa}{1+2\kappa}$, $\gamma = \frac{\kappa}{1+\kappa}$, and $\kappa = \frac{s}{1+(n-3)s}$. In particular, when $s \to 0$ (a.k.a. log-concave), $K$ is an absolute constant.*

*Proof.* Denote by $p$ the probability of observing a positive example. We discuss the following three cases.

**1. $r < m_2$ and $p < \epsilon$.**

In this case, the hypothesis that labels every examples as negative has error less than $\epsilon$. Therefore, the algorithm behaves with error at most $\epsilon$ when $r < m_2$.

**2. $r < m_2$ and $p \geq \epsilon$.**

By the Hoeffding inequality,

$$\Pr(r < m_2) \leq \Pr\left(\frac{r}{m_3} < \frac{\epsilon}{2}\right) \leq \Pr\left(\frac{r}{m_3} < p - \frac{\epsilon}{2}\right) \leq e^{-m_3\epsilon^2/2} \leq \delta/4.$$

So the probability that this case happens is at most $\delta/4$.

**3. $r \geq m_2$.**

We note that

$$\mathsf{err}(h) = \Pr(-H')\Pr(H_u \cap H_v| - H') + \Pr(H')\Pr(h_{xor}(x) \neq c(x)|x \in H'), \qquad (8)$$

where $c : \mathbb{R}^n \to \{-1, 1\}$ is the hypothesis w.r.t. $H_u \cap H_v$. Observe that

$$\Pr(-H')\Pr(H_u \cap H_v| - H') = \Pr(H_u \cap H_v)\Pr(-H'|H_u \cap H_v),$$

where $\Pr(-H'|H_u \cap H_v)$ is the error of $H'$ over the distribution conditioned on $H_u \cap H_v$. Since the VC argument works for any distribution, and $H'$ contains all $r \geq m_2$ positive examples according to Step 5 in Algorithm 4, by the VC argument, with probability at least $1 - \delta/4$,

$$\Pr(-H'|H_u \cap H_v) \leq \max\left\{\frac{\delta}{4(1+\gamma)^{1/\gamma}Km_1}, \frac{\epsilon}{2}\right\}.$$

So $\Pr(-H')\Pr(H_u \cap H_v| - H') = \Pr(H_u \cap H_v \cap (-H')) \leq \Pr(-H'|H_u \cap H_v) \leq \max\left\{\frac{\delta}{4(1+\gamma)^{1/\gamma}Km_1}, \frac{\epsilon}{2}\right\} \leq \frac{\epsilon}{2}.$

We now bound the second term in (8). According to Lemma 28,

$$\Pr((-H_u) \cap (-H_v) \cap H') \leq K\Pr(H_u \cap H_v \cap (-H')) \leq \frac{\delta}{4(1+\gamma)^{1/\gamma}m_1}.$$

On the other hand, by Lemma 8, $\Pr(H') \geq (1+\gamma)^{-1/\gamma}$ with $\gamma = s/(1+ns)$. Thus

$$\Pr((-H_u) \cap (-H_v)|H') = \frac{\Pr((-H_u) \cap (-H_v) \cap (H'))}{\Pr(H')} \leq \frac{\delta}{4m_1}.$$

That is to say, each point in $S$ has probability at most $\delta/(4m_1)$ of being in $(-H_u) \cap (-H_v)$. So by the union bound, with probability at least $1 - \delta/4$, none of points in $S$ is in $(-H_u) \cap (-H_v)$. Therefore, Step 6 in Algorithm 4 is able to find $h_{xor}$ that is consistent with all the instances in $S$. Then by the VC argument, we have

$$\Pr(h_{xor(x)} \neq c(x)|x \in H') \leq \frac{\epsilon}{2},$$

with probability at least $1 - \delta/4$. In summary, we have

$$\mathsf{err}(h) = \Pr(-H')\Pr(H_u \cap H_v| - H') + \Pr(H')\Pr(h_{xor}(x) \neq c(x)|x \in H')$$
$$\leq \frac{\epsilon}{2} + \frac{\epsilon}{2} = \epsilon,$$

with failure probability at most $\delta/4 + \delta/4 + \delta = \delta$ by the union bound. Therefore, the proof is completed. $\qquad\square$

## R  Proof of Lower Bounds

The proof of our lower bounds essentially depends on a lower bound on the packing number of all homogeneous linear separators $\mathbb{C}$ under distribution $\mathcal{D}$. Remind that the $\epsilon$-packing number, denoted by $M_{\mathcal{D}}(\mathbb{C}, \epsilon)$, is the maximal cardinality of an $\epsilon$-separated set with classifiers from $\mathbb{C}$, where we say $N$ classifiers $w_1, ..., w_N$ are $\epsilon$-separated w.r.t. $\mathcal{D}$ if $d_{\mathcal{D}}(w_i, w_j) \triangleq \Pr_{x \sim \mathcal{D}}[\mathsf{sign}(w_i \cdot x) \neq \mathsf{sign}(w_j \cdot x)] > \epsilon$ for any $i \neq j$.

**Lemma 29.** *Suppose that $\mathcal{D}$ is $s$-concave in $\mathbb{R}^n$, and that its covariance matrix is of full rank. Then for all sufficiently small $\epsilon$, we have $M_{\mathcal{D}}(\mathbb{C}, \epsilon) \geq \frac{\sqrt{n}}{2}\left(\frac{f_1(s,n)}{2\epsilon}\right)^{n-1} - 1$.*

*Proof.* We begin with proving the lemma in the case of isotropic $\mathcal{D}$. Our proof inspires from proofs for the special case of uniform and log-concave distributions by [48] and [9], respectively.

Denote by $\mathsf{UBALL}_n$ the uniform distribution on the sphere in $\mathbb{R}^n$. According to Theorem 12, for any two unit vectors $u$ and $v$ in $\mathbb{R}^n$ we have $f_1(s, n)\theta(u, v) \leq d_{\mathcal{D}}(u, v)$. Thus for a fixed $u$ the probability that a uniformly chosen $v$ obeys $d_{\mathcal{D}}(u, v) \leq \epsilon$ is upper bounded by the volume of those points in the interior of unit ball whose angle is at most $\epsilon/f_1(s, n)$ divided by the volume of unit ball in $\mathbb{R}^n$. By known bound on this ratio [48], we have $\Pr_{v \in \mathsf{UBALL}_n}[d_{\mathcal{D}}(u, v) \leq \epsilon] \leq \frac{1}{\sqrt{n}}\left(\frac{2\epsilon}{f_1(s,n)}\right)^{n-1}$. So $\Pr_{u,v \in \mathsf{UBALL}_n}[d_{\mathcal{D}}(u, v) \leq \epsilon] \leq \frac{1}{\sqrt{n}}\left(\frac{2\epsilon}{f_1(s,n)}\right)^{n-1}$, meaning that if we select a set $S$ of $s$ normalized vectors uniformly from the unit sphere, the expected number of pairs of vectors that are $\epsilon$-close in the sense of $d_{\mathcal{D}}$ is at most $\frac{s^2}{\sqrt{n}}\left(\frac{2\epsilon}{f_1(s,n)}\right)^{n-1}$. Removing one vector from each pair of $S$ yields a set of $s - \frac{s^2}{\sqrt{n}}\left(\frac{2\epsilon}{f_1(s,n)}\right)^{n-1}$ homogeneous linear separators that are $\epsilon$-separated. The proof for isotropic $\mathcal{D}$ is completed when we set $s = \frac{\sqrt{n}}{(2\epsilon/f_1(s,n))^{n-1}}$.

We now discuss the case when $\mathcal{D}$ is non-isotropic. Denote by $\Sigma$ the covariance matrix of $\mathcal{D}$ and let isotropic $\mathcal{D}'$ be the whitened version of $\mathcal{D}$, namely, the distribution obtained by first sampling $x$ from $\mathcal{D}$ and then computing $\Sigma^{-1/2}x$. Notice that $d_{\mathcal{D}}(u, v) = d_{\mathcal{D}'}(u\Sigma^{1/2}, v\Sigma^{1/2})$. Therefore, we can apply an $\epsilon$-packing w.r.t. $\mathcal{D}'$ to construct an $\epsilon$-packing w.r.t. $\mathcal{D}'$ of the same size. $\qquad\square$

Now we are ready to prove Theorem 19.

**Theorem 19** (restated) *For a fixed value $-\frac{1}{2n+3} \leq s \leq 0$ we have: (a) For any $s$-concave distribution $\mathcal{D}$ in $\mathbb{R}^n$ whose covariance matrix is of full rank, the sample complexity of learning origin-centered*

*linear separators under $\mathcal{D}$ in the passive learning scenario is $\Omega\left(\frac{nf_1(s,n)}{\epsilon}\right)$; (b) The label complexity of active learning of linear separators under s-concave distribution is $\Omega\left(n\log\left(\frac{f_1(s,n)}{\epsilon}\right)\right)$.*

*Proof.* It is known that for any distribution $\mathcal{D}$ in $\mathbb{R}^n$, the sample complexity of (passive) PAC learning of homogeneous linear separators under $\mathcal{D}$ is at least $\frac{n-1}{e}\left(\frac{M_{\mathcal{D}}(\mathbb{C},2\epsilon)}{4}\right)^{1/(n-1)}$ [48]. By Lemma 29, we have an $\Omega\left(\frac{nf_1(s,n)}{\epsilon}\right)$ lower bound of sample complexity for passive learning homogeneous halfspace.

We now discuss the label complexity lower bound in the active learning scenario. By [46], any active learning algorithm that is allowed to make arbitrary binary queries must take at least $\Omega(\log M_{\mathcal{D}}(\mathbb{C},\epsilon))$ so as to output a hypothesis of error at most $\epsilon$ with high probability. Applying Lemma 29, we obtain the desired result. $\square$

# S    Related Algorithms

## S.1    Margin Based Active Learning (Realizable Case)

---

**Algorithm 2** Margin Based Active Learning under S-Concave Distributions (Realizable Case)

---

**Input:** $b_k = \min\{\Theta(2^{-k}f_4 f_1^{-1}), d\}$, $m_k = C\left(\frac{f_3 b_{k-1}}{2^{-k}}\left(n\log\frac{f_3 b_{k-1}}{2^{-k}} + \log\frac{1+s-k}{\delta}\right)\right)$, and $T = \lceil\log\frac{1}{c\epsilon}\rceil$.
**1:** Draw $m_1$ examples from $\mathcal{D}$, label them and put them into $W(1)$.
**2: For** $k = 1,2,...,T$
**3:**    Find a hypothesis $w_k$ with $\|w_k\| = 1$ that is consistent with $W(k)$.
**4:**    $W(k+1) \leftarrow W(k)$.
**5:**    **While** $m_{k+1}$ additional data points are not labeled
**6:**       Draw sample $x$ from $\mathcal{D}$.
**7:**       **If** $|w_k \cdot x| \geq b_k$
**8:**          Reject $x$.
**9:**       **Else**
**10:**          Ask for label of $x$ and put into $W(k+1)$.
**11:**       **End If**
**12:**    **End While**
**13: End For**
**Output:** Hypothesis $w_T$.

---

## S.2    Margin Based Active Learning (Adversarial Noise)

---

**Procedure 3** Margin Based Active Learning under S-Concave Distributions (Adversarial Noise)

---

**Input:** Parameters $b_k$, $\tau_k$, $r_k$, $m_k$, $\kappa$, and $T$ as in Theorem 16.
**1:** Draw $m_1$ examples from $\mathcal{D}$, label them and put them into $W$.
**2: For** $k = 1,2,...,T$
**3:**    Find $v_k \in \mathsf{ball}(w_{k-1}, r_k)$ to approximately minimize the hinge loss over $W$ s.t. $\|v_k\| \leq 1$: $\ell_{\tau_k} \leq \min_{w\in\mathsf{ball}(w_{k-1},r_k)\cap\mathsf{ball}(0,1)}\ell_{\tau_k}(w,W) + \kappa/8$.
**4:**    Normalize $v_k$, yielding $w_k = \frac{v_k}{\|v_k\|}$.
**5:**    Clear the working set $W$.
**6:**    **While** $m_{k+1}$ additional data points are not labeled
**7:**       Draw sample $x$ from $\mathcal{D}$.
**8:**       **If** $|w_k \cdot x| \geq b_k$, reject $x$; **else** ask for label of $x$ and put into $W$.
**9:**    **End While**
**10: End For**
**Output:** Hypothesis $w_T$.

---

## S.3 Learning Intersections of Halfspaces

---

**Algorithm 4** Learning Intersections of Halfspaces under S-Concave Distributions

---

**Input:** Parameters $m_1$, $m_2$, and $m_3$ as in Theorem 18.

**1:** Draw $m_3$ examples. Denote by $r$ the number of observed positive examples.

**2: If** $r < m_2$, output the hypothesis that labels every point as negative, and end the algorithm.

**3:** Learn an origin-centered halfspace $H'$ which contains all $r$ positive examples.

**4:** Draw a set $S$ of $m_1$ i.i.d. examples in $H'$. Learn a weight vector $w \in \mathbb{R}^{n \times n}$ such that the hypothesis $h_{xor} = \text{sign}\left(\sum_{i=1}^{n}\sum_{j=1}^{n} w_{ij} x_i x_j\right)$ is consistent with the set $S$.

**Output:** $h : \mathbb{R}^n \to \{-1, 1\}$ such that $h(x) = h_{xor}(x)$ if $x \in H'$; Otherwise, $h(x) = -1$.

---

# T   A Collection of Concentration Results

**Theorem 30** ([54, 17]). *Denote by $\mathcal{C}$ a class of concepts from a set $X$ to $\{-1, 1\}$ with VC dimension $n$. Let $c \in \mathcal{C}$, and assume that*

$$M(\epsilon, \delta, n) = O\left(\frac{n}{\epsilon}\log\frac{1}{\epsilon} + \frac{1}{\epsilon}\log\frac{1}{\delta}\right)$$

*examples $x_1, ..., x_M$ are sampled from any probability distribution $\mathcal{D}$ over $X$. Then any hypothesis $h \in \mathcal{C}$ which is consistent with $c$ on $x_1, ..., x_M$ has error at most $\epsilon$, with probability at least $1 - \delta$.*

**Theorem 31** ([1]). *Let $F$ be a set of functions mapping from domain $X$ to $[a, b]$, and let $n$ be the pseudo-dimension of $F$. Then for any distribution $\mathcal{D}$ over $X$ and $m = O\left(\frac{(b-a)^2}{\kappa^2}(d + \log(1/\delta))\right)$, if $x_1, ..., x_m$ are drawn independently from $\mathcal{D}$, with probability at least $1 - \delta$, for all $f \in F$,*

$$\left|\mathbb{E}_{x\sim\mathcal{D}}f(x) - \frac{1}{m}\sum_{i=1}^{m} f(x_i)\right| \leq \kappa.$$