[Reviews · NeurIPS 2017]

Reviewer 1



This paper talks about noise tolerant learning (both active and passive) for s-concave distributions. Learning results have been established for the more restrictive log-concave distributions and this paper goes beyond log-concave distributions and provides variants of well known AL algorithms such as Margin Based Active learning (for the active learning setup) and a variant of the Baum's algorithm (for passive learning setup). While the algorithms themselves are not new (in stylistic sense), a lot of technical results regarding s-concave distributions needs to be derived that can be used in the analysis and design of the above mentioned algorithms. The central contribution is essentially deriving these convex geometric results. I do not have any major objective comments, except about the importance of this work. My main reservation is that while the paper seems to advance the state-of-art in learning theory, I am not sure if these algorithms and results will have much impact outside the learning theory community.

Reviewer 2



This paper extends several previous results on active learning and efficient passive learning from isotropic log-concave distributions to isotropic s-concave distributions with a certain range of s values. The latter is a more general class, that includes log-concave distributions. The results are derived from several new geometric bounds on such distributions, which are similar to previous bounds for log-concave distributions, but more general, capturing also the fact that s-concave distributions can have fat tails. All of the proofs are provided in the appendix, which I did not verify. The algorithms used are the same ones previously used for log-concave distributions. The results presented in the paper are quite technical. While the paper is relatively clearly written, the results themselves are presented in a highly technical manner which is not easy to interpret. For instance, Theorem 15 and 16 give very complicated label complexity bounds for a single algorithmic iteration. They do not give a single, interpretable, label complexity bound which holds for the entire algorithm. I recommend that the authors improve this aspect of their presentation. Overall, the paper presents results that advance the theory of active and passive learning. However, due to the technicality of the results and the presentation, it is hard to gain scientific insight from them.

Reviewer 3



This review is adapted from my review from COLT 2017 - My feedback to this paper has not changed much since then. This paper studies a new family of distributions, s-concave distributions, which appears in works of (Brascamp and Lieb, 1976; Bobkov, 2007, Chandrasekaran, Deshpande and Vempala, 2009). The main result is a series of upper and lower bounds regarding its probability distribution function and its measure over certain regions. These inequalities can be readily applied to (active) learning linear separators and learning the intersection of two halfspaces. Overall this is an interesting paper, extending the family of distributions in which the problem of learning linear separators can be efficiently solved. This may spur future research on establishing new distribution-dependent conditions for (efficient) learnability. Technical quality: on one hand, the results are impressive, since the family of admissible distributions (in the sense of Awasthi, Balcan and Long, 2014 and Klivans, Long and Tang, 2009) is broadened; on the other hand, the analysis only works when -1/(2n+3)<=s<=0. In high dimensional settings, the range of s is fairly small. Novelty: the result of this paper follows the reasoning in (Lovasz and Vempala, 2007); but this is definitely a non-trivial extension. Potential Impact: this work may spur future research on distributional conditions for efficient learnability.